# Apollo-MILP: An Alternating Prediction-Correction Neural Solving Framework for Mixed-Integer Linear Programming

**Haoyang Liu**[1],[*] **Jie Wang**[1],[†] **Zijie Geng**[1], **Xijun Li**[4,2], **Yuxuan Zong**[1], **Fangzhou Zhu**[2], **JianYe Hao**[2,3], **Feng Wu**[1]

[1]MoE Key Laboratory of Brain-inspired Intelligent Perception and Cognition, University of Science and Technology of China
[2] Noah's Ark Lab, Huawei Technologies
[3] Tianjin University
[4] Shanghai Jiao Tong University

## Abstract

Leveraging machine learning (ML) to predict an initial solution for mixed-integer linear programming (MILP) has gained considerable popularity in recent years. These methods predict a solution and fix a subset of variables to reduce the problem dimension. Then, they solve the reduced problem to obtain the final solutions. However, directly fixing variable values can lead to low-quality solutions or even infeasible reduced problems if the predicted solution is not accurate enough. To address this challenge, we propose an Alternating prediction-correction neural solving framework (Apollo-MILP) that can identify and select accurate and reliable predicted values to fix. In each iteration, Apollo-MILP conducts a prediction step for the unfixed variables, followed by a correction step to obtain an improved solution (called reference solution) through a trust-region search. By incorporating the predicted and reference solutions, we introduce a novel Uncertainty-based Error upper BOund (UEBO) to evaluate the uncertainty of the predicted values and fix those with high confidence. A notable feature of Apollo-MILP is the superior ability for problem reduction while preserving optimality, leading to high-quality final solutions. Experiments on commonly used benchmarks demonstrate that our proposed Apollo-MILP significantly outperforms other ML-based approaches in terms of solution quality, achieving over a 50% reduction in the solution gap.

## 1 Introduction

Mixed-integer linear programming (MILP) is one of the most fundamental models for combinatorial optimization with broad applications in operations research (Bixby et al., 2004), engineering (Ma et al., 2019), and daily scheduling or planning (Li et al., 2024b). However, solving large-size MILPs remains time-consuming and computationally expensive, as many are NP-hard and have exponential expansion of search spaces as instance sizes grow. To mitigate this challenge, researchers have explored a wide suite of machine learning (ML) methods (Gasse et al., 2022). In practice, MILP instances from the same scenario often share similar patterns and structures, which ML models can capture to achieve improved performance (Bengio et al., 2021).

Recently, extensive research has focused on using ML models to predict solutions for MILPs. Notable approaches include Neural Diving (ND) (Nair et al., 2020; Yoon, 2021; Paulus & Krause, 2023) and Predict-and-Search (PS) (Han et al., 2023; Huang et al., 2024), as illustrated in Figure 1. Given a MILP instance, ND and PS begin by employing an ML model to predict an initial solution. ND with SelectiveNet (Nair et al., 2020) assigns fixed values to a subset of variables based on the prediction, thereby constructing a reduced MILP problem with a reduced dimensionality of decision variables. Then, ND solves the reduced problem to obtain the final solutions. However, the fixing strategy

---

[*]This work was done when Haoyang Liu was an intern at Huawei.
[†]Corresponding author. Email: jiewangx@ustc.edu.cn.

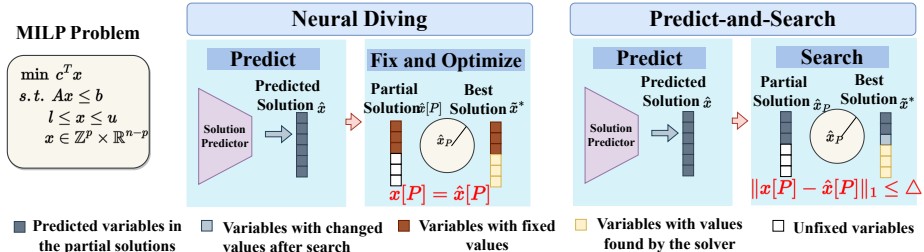

Figure 1: Illustration of Neural Diving (ND) and Predict-and-Search (PS). For a given MILP problem, both methods begin by using a GNN predictor to generate an initial solution $\hat{x}$ and construct a partial solution $\hat{x}[P]$. ND then fixes the variable values in this partial solution and optimizes the reduced problem. While PS searches within a neighborhood around the partial solution.

faces several limitations. The solving efficiency and the quality of the final solutions heavily depend on the accuracy of the ML-based predictor for initial solutions (Huang et al., 2024), but achieving an accurate ML-based predictor is often challenging due to the complex combinatorial nature of MILPs, insufficient training data, and limited model capacity. Consequently, enforcing variables to fixed values that may not be accurate can misguide the search toward areas that do not contain the optimal solution, leading to low-quality final solutions or even infeasible reduced problems. Instead of fixing variables, PS offers a more effective search strategy for a pre-defined neighborhood of the predicted partial solution, leading to better feasibility and higher-quality final solutions. The trust-region search strategy in PS allows better feasibility but is less effective than the fixing strategy in terms of problem dimension reduction, as it requires a larger search space.

To address the aforementioned challenges, a natural idea is to refine the predicted solutions before fixing them. We observe that the search process in PS provides valuable feedback to enhance solution quality for prediction, an aspect that has been overlooked in existing research. Specifically, the solver guides the searching direction toward the optimal solution while correcting variable values that are inappropriately fixed. Theoretically, incorporating this correction yields higher precision for predicted solutions (please see Theorem 3).

In light of this, we propose a novel MILP optimization approach, called the Alternating Prediction-Correction Neural Solving Framework (Apollo-MILP), that can effectively identify the correct and reliable predicted variable values to fix. In each iteration, Apollo-MILP conducts a prediction step for the unfixed variables, followed by a correction step to obtain an improved solution (called reference solution) through a trust-region search. The reference solution serves as guidance provided by the solver to correct the predicted solution. By incorporating both predicted and reference solutions, we introduce a novel Uncertainty-based Error upper BOund (UEBO) to evaluate the uncertainty of the predicted values and fix those with high confidence. Furthermore, we also propose a straightforward variable fixing strategy based on UEBO. Theoretical results show that this strategy guarantees improved solution quality and feasibility for the reduced problem. Experiments demonstrate that Apollo-MILP reduces the solution gap by over 50% across various popular benchmarks, while also achieving higher-quality solutions in one-third of the runtime compared to traditional solvers.

We highlight our main contributions as follows. (1) A Novel Prediction-Correction MILP Solving Framework. Apollo-MILP is the first framework to incorporate a correction mechanism to enhance the precision of solution predictions, enabling effective problem reduction while preserving optimality. (2) Investigating Effective Problem Reduction Techniques. We rethink the existing problem-reduction techniques for MILPs and establish a comprehensive criterion for selecting an appropriate subset of variable values to fix, combining the advantages of existing search and fixing strategies. (3) High Performance across Various Benchmarks. We conduct extensive experiments demonstrating Apollo-MILP's strong performance, generalization ability, and real-world applicability.

## 2    RELATED WORKS

**ML-Enhanced Branch-and-Bound Solver**    In practice, typical MILP solvers, such as SCIP (Achterberg, 2009) and Gurobi (Gurobi Optimization, 2021), are primarily based on the Branch-and-Bound (B&B) algorithm. ML has been successfully integrated to enhance the solving efficiency of

these B&B solvers (Bengio et al., 2021; Li et al., 2024a; Gasse et al., 2022; Scavuzzo et al., 2024). Specifically, many researchers have leveraged advanced techniques from imitation and reinforcement learning to improve key heuristic modules. A significant portion of this work aims to learn heuristic policies for selecting variables to branch on (Khalil et al., 2016; Gasse et al., 2019; Gupta et al., 2020; Zarpellon et al., 2021; Gupta et al., 2022; Scavuzzo et al., 2022; Lin et al., 2024; Zhang et al., 2024; Kuang et al., 2024), selecting cutting planes (Tang et al., 2020; Wang et al., 2023b; 2024b; Huang et al., 2022; Balcan et al., 2022; Paulus et al., 2022; Ling et al., 2024; Puigdemont et al., 2024), and determining which nodes to explore next (He et al., 2014; Labassi et al., 2022; Liu et al., 2024c). These ML-enhanced methods have demonstrated substantial improvements in solving efficiency. Additionally, extensive research has been dedicated to boosting other critical modules in the B&B algorithm, such as separation (Li et al., 2023), scheduling of primal heuristics (Khalil et al., 2017; Chmiela et al., 2021), presolving (Liu et al., 2024a), data generation (Geng et al., 2023; Liu et al., 2023a; 2024b) and large neighborhood search (Song et al., 2020; Wu et al., 2021; Sonnerat et al., 2021; Huang et al., 2023). Beyond practical applications, theoretical advancements have also emerged to analyze the expressiveness of GNNs for MILPs and LPs (Chen et al., 2023a;b), as well as to develop landscape surrogates for ML-based solvers (Zharmagambetov et al., 2023).

**ML for Solution Prediction**  Another line of research leverages ML models to directly predict solutions (Ding et al., 2020; Yoon, 2021; Khalil et al., 2022; Paulus & Krause, 2023; Zeng et al., 2024; Cai et al., 2024; Li et al., 2025; Geng et al., 2025). Neural Diving (ND) Nair et al. (2020) is a pioneering approach in this field. Specifically, ND predicts a partial solution based on coverage rates and utilizes SelectiveNet to determine which predicted variables to fix. To enhance the quality of the final solution, subsequent methods incorporate search mechanisms, such as trust-region search (PS) Han et al. (2023); Huang et al. (2024) and large neighborhood search Sonnerat et al. (2021); Ye et al. (2023; 2024) with sophisticated neighborhood optimization techniques. In this paper, we focus on ND and PS, both of which have gained significant popularity in recent years.

## 3 PRELIMINARIES

### 3.1 MIXED INTEGER LINEAR PROGRAMMING

A mixed-integer linear programming (MILP) is defined as follows,

$$\min_{\boldsymbol{x} \in \mathbb{R}^n} \quad \boldsymbol{c}^\top \boldsymbol{x}, \quad \text{s.t.} \quad \boldsymbol{A}\boldsymbol{x} \leq \boldsymbol{b}, \boldsymbol{l} \leq \boldsymbol{x} \leq \boldsymbol{u}, \boldsymbol{x} \in \mathbb{Z}^p \times \mathbb{R}^{n-p}, \tag{1}$$

where $\boldsymbol{x}$ denotes the $n$-dimensional decision variables, consisting of $p$ integer components and $n-p$ continuous variables. The vector $\boldsymbol{c} \in \mathbb{R}^n$ denotes the coefficients of the objective function, $\boldsymbol{A} \in \mathbb{R}^{m \times n}$ is the constraint coefficient matrix, and $\boldsymbol{b} \in \mathbb{R}^m$ represents the right-hand side terms of the constraints. The vectors $\boldsymbol{l} \in (\mathbb{R} \cup \{-\infty\})^n$ and $\boldsymbol{u} \in (\mathbb{R} \cup \{+\infty\})^n$ specify the lower and upper bounds for the variables, respectively. It is reasonable that PS primarily focuses on mixed-binary programming with $\boldsymbol{x} \in \{0, 1\}^p \times \mathbb{R}^{n-p}$ for simplification, as it can be easily generalized to general MILPs using the well-established modification techniques proposed in Nair et al. (2020).

### 3.2 BIPARTITE GRAPH REPRESENTATION FOR MILPS

A MILP instance can be represented as a weighted bipartite graph $\mathcal{G} = (\mathcal{W} \cup \mathcal{V}, \mathcal{E})$ Gasse et al. (2019), as illustrated in Figure 2. In this bipartite graph, the two sets of nodes, $\mathcal{W}$ and $\mathcal{V}$, represent the constraints and variables in the MILP instance, respectively. An edge is constructed between a constraint node and a variable node if the variable has a nonzero coefficient in the constraint. For further details on the graph features utilized in this paper, please refer to Appendix E.

### 3.3 PREDICT-AND-SEARCH

Predict-and-Search (PS) Han et al. (2023) is a two-stage MILP optimization framework that utilizes machine learning models to learn the Bernoulli distribution for the solution values of binary variables. It then performs a trust-region search within a neighborhood of the predicted solution $\hat{\boldsymbol{x}}$ to enhance solution quality. Given a MILP instance $\mathcal{I}$, PS considers approximating the solution

distribution $q(\boldsymbol{x} \mid \mathcal{I})$ by weighing the solutions with their objective value,

$$q(\boldsymbol{x} \mid \mathcal{I}) = \frac{\exp(-E(\boldsymbol{x}, \mathcal{I}))}{\sum_{\boldsymbol{x}' \in \mathcal{S}} \exp(-E(\boldsymbol{x}', \mathcal{I}))}, \text{ where the energy function } E(\boldsymbol{x}, \mathcal{I}) = \begin{cases} \boldsymbol{c}^\top \boldsymbol{x}, & \text{if } \boldsymbol{x} \text{ is feasible,} \\ +\infty, & \text{otherwise,} \end{cases}$$

and $\mathcal{S}$ is a collected set of optimal or near-optimal solutions. PS learns the solution distribution using a GNN model $p_\theta$ and computes the marginal probability $p_\theta(\boldsymbol{x} \mid \mathcal{I})$ to predict a solution. To simplify the formulation, PS assumes that the variables are independent, as described in Nair et al. (2020), i.e., $p_\theta(\boldsymbol{x} \mid \mathcal{I}) = \prod_{i=1}^n p_\theta(\boldsymbol{x}_i \mid \mathcal{I})$. PS then selects $k_1$ binary variables with the highest predicted marginal values $p_\theta(\boldsymbol{x}_i \mid \mathcal{I})$ and fixes them to 1. Similarly, PS fixes $k_0$ binary variables with the lowest marginal values to 0. The hyperparameters $k_0$ and $k_1$ are called partial solution size parameters, and we denote the fixed partial solution as $\hat{\boldsymbol{x}}[P]$, where $P$ is the index set for the fixed variables with $k_0 + k_1$ elements. Instead of directly fixing the variables $\hat{\boldsymbol{x}}[P]$, PS employs a traditional solver, such as SCIP or Gurobi, to explore the neighborhood $\mathcal{B}_P(\hat{\boldsymbol{x}}[P], \triangle)$ of the predicted partial solution $\hat{\boldsymbol{x}}[P]$ in search of the best feasible solution. Here $\triangle$ represents the trust-region radius (neighborhood parameter), and $\mathcal{B}_P(\hat{\boldsymbol{x}}[P], \triangle) = \{\boldsymbol{x}[P] \in \mathbb{R}^n \mid \|\hat{\boldsymbol{x}}[P] - \boldsymbol{x}[P]\|_1 \leq \triangle\}$ is the trust region. The neighborhood search process is formulated as the following MILP problem, referred to as the *trust-region search problem*,

$$\min_{\boldsymbol{x} \in \mathbb{R}^n} \quad \boldsymbol{c}^\top \boldsymbol{x}, \quad \text{s.t.} \quad \boldsymbol{A}\boldsymbol{x} \leq \boldsymbol{b}, \boldsymbol{l} \leq \boldsymbol{x} \leq \boldsymbol{u},$$
$$\boldsymbol{x}[P] \in \mathcal{B}_P(\hat{\boldsymbol{x}}[P], \triangle), \boldsymbol{x} \in \mathbb{Z}^p \times \mathbb{R}^{n-p}. \tag{2}$$

Notice that PS reduces to ND (without SelectiveNet) when the neighborhood parameter $\triangle = 0$.

## 4    THE PROPOSED ALTERNATING PREDICTION-CORRECTION FRAMEWORK

"*How to identify and fix a high-quality partial solution*" is a longstanding challenge for ML-based solution prediction approaches. Unlike existing works that primarily focus on enhancing the prediction accuracy of ML models, we offer a new perspective by identifying and selecting the correct and reliable predicted values to fix, thereby improving the quality of final solutions and overall solving efficiency. The proposed Apollo-MILP framework alternates between prediction (Section 4.1) and correction (Section 4.2) steps, progressively identifying high-confidence variables and expanding the subset of fixed variables. As the algorithm proceeds, we obtain a sequence of MILPs $\mathcal{I}^{(0)} \rightarrow \mathcal{I}^{(1)} \rightarrow \cdots \rightarrow \mathcal{I}^{(K)}$ with fewer decision variables, where the superscripts $\{(k) \mid k = 0, \cdots, K\}$ is the iteration number. The overview of our architecture is in Figure 2.

As shown in Figure 2, during the $k^{\text{th}}$ iteration, Apollo-MILP processes a MILP $\mathcal{I}^{(k)}$, which may be either the original problem or a reduced version. In this section, we assume that the prediction and correction steps are performed within the $k^{\text{th}}$ iteration. Therefore, we simplify the notation by omitting the superscript $(k)$ without leading to misunderstanding. For example, we denote the predicted solution by $\hat{\boldsymbol{x}}$ instead of $\hat{\boldsymbol{x}}^{(k)}$.

### 4.1    PREDICTION STEP

In each prediction step, our goal is to predict the solution for the current MILP problem $\mathcal{I}$, which is represented as a bipartite graph, as discussed in Section 3.2.

We employ a GNN-based solution predictor $p_\theta$ to predict the marginal probabilities of values $p_\theta(\boldsymbol{x} \mid \mathcal{I})$ for binary variables in the optimal solution, similar to the method employed in PS (Han et al., 2023). Assuming independence among the variables, the predictor outputs the probability that the variable equals 1, i.e., $p_\theta(\boldsymbol{x}_i = 1 \mid \mathcal{I})$ for $i = 1, \cdots, n$.

As mentioned above, the predictor takes either the original or reduced MILP problems as input. However, the distribution of the reduced problems may differ from that of the original problems. To address this issue, we employ data augmentation to align the distributional shifts. Specifically, for a given MILP instance $\mathcal{I}$ in the training dataset, we collect a set $\mathcal{S}_\mathcal{I}$ of $m$ optimal or near-optimal solutions to approximate the solution distribution $q(\boldsymbol{x} \mid \mathcal{I})$ mentioned in Section 3.3. We then randomly sample a solution $\boldsymbol{x}^*$ from this solution pool $\mathcal{S}_\mathcal{I}$, along with a subset of variables from $\mathcal{I}$. We fix the selected variables to the corresponding values in $\boldsymbol{x}^*$ to generate a reduced instance $\mathcal{I}'$.

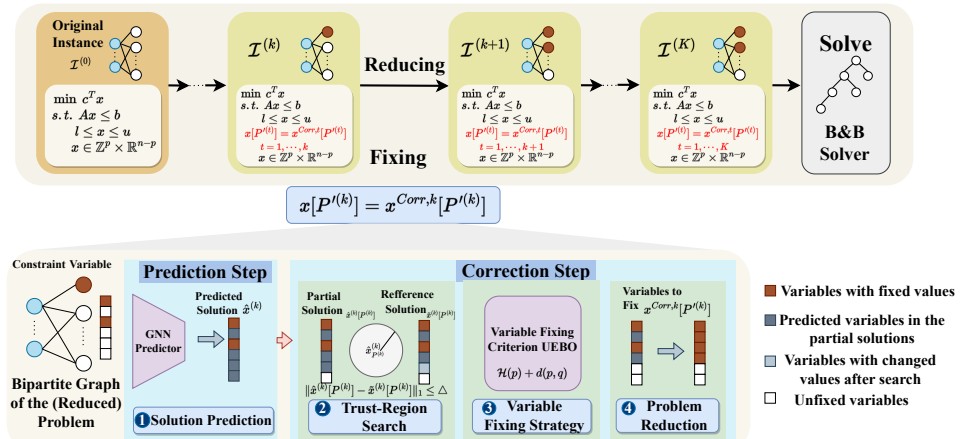

Figure 2: The overview of Apollo-MILP. Apollo-MILP operates through an iterative process that alternates between prediction and correction steps to reduce the original MILP problem progressively. In the prediction step, Apollo-MILP (1) employs a GNN to generate a partial solution. In the correction step, (2) a trust region-based search is conducted to refine this solution to obtain the reference solution. (3) The proposed variable fixing criterion, UEBO, is then calculated to identify which variables should be fixed. (4) Finally, we reduce the problem dimension by enforcing the selected variable values to fix values.

For each reduced instance $\mathcal{I}'$, we also collect $m$ optimal or near-optimal solutions to estimate the solution distribution $q(\boldsymbol{x} \mid \mathcal{I}')$. All instances and solutions are combined, resulting in an enriched training dataset denoted as $\mathcal{D}$.

To calculate the prediction target for training, we construct the estimated probability target vector $(\boldsymbol{p}(\boldsymbol{x}_1 = 1 \mid \mathcal{I}), \boldsymbol{p}(\boldsymbol{x}_2 = 1 \mid \mathcal{I}), \cdots, \boldsymbol{p}(\boldsymbol{x}_n = 1 \mid \mathcal{I}))^\top$. Here, we let

$$\boldsymbol{p}_i = \boldsymbol{p}(\boldsymbol{x}_i = 1 \mid \mathcal{I}) = \frac{\sum_{\boldsymbol{x}' \in \mathcal{S}_\mathcal{I}, x_i' = 1} \exp(-\boldsymbol{c}^\top \boldsymbol{x}')}{\sum_{\boldsymbol{x}' \in \mathcal{S}_\mathcal{I}} \exp(-\boldsymbol{c}^\top \boldsymbol{x}')} \tag{3}$$

be the probability of variable $\boldsymbol{x}_i$ being assigned the value 1, given the instance $\mathcal{I}$ from the enriched dataset $\mathcal{D}$ and the solution set $\mathcal{S}_\mathcal{I}$. Finally, the predictor $p_\theta$ is trained by minimizing the cross-entropy loss (Han et al., 2023)

$$\mathcal{L}(\theta) = -\frac{1}{|\mathcal{D}|} \sum_{(\mathcal{I}, \mathcal{S}_\mathcal{I}) \in \mathcal{D}} \sum_{i=1}^{n} \left(\boldsymbol{p}_i \log p_\theta(\boldsymbol{x}_i = 1 \mid \mathcal{I}) + (1 - \boldsymbol{p}_i) \log(1 - p_\theta(\boldsymbol{x}_i = 1 \mid \mathcal{I}))\right). \tag{4}$$

## 4.2 CORRECTION STEP

The correction step aims to improve the partial solutions by identifying and discarding the inaccurate predicted variable values that were inappropriately fixed. Specifically, we (1) leverage a trust-region search on the partial solution for a refined solution as a reference for subsequent operations, (2) introduce a novel uncertainty-based metric to determine which subset of variables to fix, and (3) enforce the selected variables to fixed values for dimension reduction.

To begin with, we establish the following notations. Given a MILP instance $\mathcal{I}$, let $q(\boldsymbol{x} \mid \mathcal{I})$ represent the distribution of the optimal solution, and $q(\boldsymbol{x} \mid \hat{\boldsymbol{x}}, \mathcal{I})$ denote the distribution of the reference solution given instance $\mathcal{I}$ and predicted solution $\hat{\boldsymbol{x}}$. The notation $\boldsymbol{x}[P]$ implies that the partial solution $\boldsymbol{x}[P]$ has the same variable values as $\boldsymbol{x}$ in the index set $P$.

**Trust-Region Search** We leverage the solver as a corrector to improve the predicted solutions via trust-region search. Formally, given the predicted marginal probabilities $p_\theta(\boldsymbol{x} \mid \mathcal{I})$, we solve the MILP problem 2 with predefined hyperparameters $(k_0, k_1, \triangle)$, which is similar to the search process in PS. In this process, the partial solution to be fixed during the search is $\hat{\boldsymbol{x}}[P]$. The best primal solution $\tilde{\boldsymbol{x}} \sim q(\boldsymbol{x} \mid \hat{\boldsymbol{x}}, \mathcal{I})$ found by the solver has values $\tilde{\boldsymbol{x}}[P]$ for the variable index set $P$.

**Correction Criterion**   The solution $\tilde{\boldsymbol{x}}$ obtained through the trust-region search serves as a reference to improve the solution quality, called the reference solution. Then, we need to determine which variables to fix and the values they should be assigned. To evaluate the reliability of the predictions for each variable, a natural approach is to compute the distributional discrepancy between the optimal and predicted solutions, specifically $D_{\mathrm{KL}}\left(p_{\boldsymbol{\theta}}(\boldsymbol{x}_i \mid \mathcal{I})\|q(\boldsymbol{x}_i \mid \mathcal{I})\right)$, where we have assumed the independence between different variables. Here the (conditional) Kullback–Leibler (KL) divergence is defined to be $D_{KL}(p\|q) = \sum_k p(y_k) \log \frac{p(y_k)}{q(y_k)}$ for distributions $p, q$ and variable $y$ taking values in $\{y_1, y_2, \cdots, y_k, \cdots\}$. However, during testing, the optimal solution is not available, rendering the computation of the KL divergence intractable. Fortunately, we propose an upper bound to estimate the KL divergence that utilizes the available reference solutions.

**Proposition 1** (Uncertainty-Based Error Upper Bound). *We derive the following upper bound for the KL divergence between the predicted marginal probability $p_{\boldsymbol{\theta}}(\boldsymbol{x}_i \mid \mathcal{I})$ and optimal solution distribution $q(\boldsymbol{x}_i \mid \mathcal{I})$, utilizing $p_{\boldsymbol{\theta}}(\boldsymbol{x}_i \mid \mathcal{I})$ and the reference solution distribution $q(\boldsymbol{x}_i \mid \hat{\boldsymbol{x}}_i, \mathcal{I})$,*

$$\underbrace{D_{KL}\left(p_{\boldsymbol{\theta}}(\boldsymbol{x}_i \mid \mathcal{I})\|q(\boldsymbol{x}_i \mid \mathcal{I})\right)}_{\text{Target Distance}} \leq \underbrace{\mathcal{H}(p_{\boldsymbol{\theta}}(\boldsymbol{x}_i \mid \mathcal{I}))}_{\text{Prediction Uncertainty}} + \underbrace{d(p_{\boldsymbol{\theta}}(\boldsymbol{x}_i \mid \mathcal{I}), q(\boldsymbol{x}_i \mid \hat{\boldsymbol{x}}_i, \mathcal{I}))}_{\text{Prediction-Correction Discrepancy}}, \tag{5}$$

*where $\mathcal{H}(\cdot)$ denotes the entropy with $\mathcal{H}(p) = -\sum_k p(y_k)log(p(y_k))$ for variable $y$ taking values in $\{y_1, y_2, \cdots, y_k, \cdots\}$, and $d(\cdot, \cdot)$ represents the (conditional) cross-entropy loss of distributions with $d(p, q) = -\sum_k q(y_k) \log(p(y_k))$.*

We define the upper bound in Equation (5) as the Uncertainty-based Error upper BOund (UEBO), represented as $\mathrm{UEBO}(p, q) := \mathcal{H}(p) + d(p, q)$ for distributions $p$ and $q$. The first term $\mathcal{H}(p_{\boldsymbol{\theta}}(\boldsymbol{x}_i \mid \mathcal{I}))$ on the right-hand side of Equation (5), referred to as prediction uncertainty, reflects the confidence of the predictor in its predictions. A lower negative entropy value indicates lower uncertainty and greater confidence in the predictor $p_{\boldsymbol{\theta}}$. The second term $d(p_{\boldsymbol{\theta}}(\boldsymbol{x}_i \mid \mathcal{I}), q(\boldsymbol{x}_i \mid \hat{\boldsymbol{x}}_i, \mathcal{I}))$, called the prediction-correction discrepancy, quantifies the divergence between the predicted and reference solutions. A larger discrepancy suggests that further scrutiny of the predicted results is necessary. We will now discuss why UEBO has the potential to be an effective metric for selecting which variables to fix.

1. **Providing an upper bound** of the intractable KL divergence $D_{\mathrm{KL}}\left(p_{\boldsymbol{\theta}}(\boldsymbol{x}_i \mid \mathcal{I})\|q(\boldsymbol{x}_i \mid \mathcal{I})\right)$. During testing, the distribution of the optimal solution $q(\boldsymbol{x}_i \mid \mathcal{I})$ is generally unknown, making the computation of this KL divergence intractable. Instead, UEBO offers a practical estimation by utilizing the available distributions $p_{\boldsymbol{\theta}}(\boldsymbol{x}_i \mid \mathcal{I}), q(\boldsymbol{x}_i \mid \hat{\boldsymbol{x}}_i, \mathcal{I})$.

2. **Estimating the discrepancy** between the solutions. UEBO aims to penalize variables that exhibit substantial prediction uncertainty or significant disagreement, indicating the reliability of the predicted values.

**Problem Reduction.**   We begin by selecting variables with low UEBO according to the correction rule, as low UEBO indicates higher reliability and greater potential for high-quality solutions. Consequently, we can be more confident in fixing these variables to construct a partial solution $\boldsymbol{x}^{Corr}[P'] = \mathcal{F}(\hat{\boldsymbol{x}}[P], \tilde{\boldsymbol{x}}[P])$, referred to as the corrected partial solution. Specifically, the correction operator $\mathcal{F}$ takes in the predicted and reference partial solutions $\hat{\boldsymbol{x}}[P]$ and $\tilde{\boldsymbol{x}}[P]$ and identifies a new index set $P' \subset P$ of variables to fix, along with their corresponding fixed values. Finally, we arrive at the following reduced problem for the next iteration.

$$\begin{aligned} \min_{\boldsymbol{x} \in \mathbb{R}^n} \quad & \boldsymbol{c}^\top \boldsymbol{x}, \quad \text{s.t.} \quad \boldsymbol{A}\boldsymbol{x} \leq \boldsymbol{b}, \boldsymbol{l} \leq \boldsymbol{x} \leq \boldsymbol{u}, \\ & \boldsymbol{x}[P] = \boldsymbol{x}^{Corr}[P'], \quad \boldsymbol{x} \in \mathbb{Z}^p \times \mathbb{R}^{n-p}. \end{aligned} \tag{6}$$

Furthermore, to accelerate the convergence, we can introduce a cut $\boldsymbol{c}^\top \boldsymbol{x} < \boldsymbol{c}^\top \tilde{\boldsymbol{x}}$ into the reduced problem to ensure monotonic improvement.

### 4.3   ANALYSIS OF THE FIXING STRATEGY

This part is organized as follows. (1) We begin by introducing the concept of prediction-correction consistency for a variable and illustrating its close relationship with UEBO. (2) We propose a straightforward strategy $\mathcal{F}$ for approximating UEBO and fixing variables. (3) We analyze the advancement properties of Apollo-MILP incorporated with the proposed fixing strategy.

**(1) UEBO and Prediction-Correction Consistency**   To provide deeper insight into UEBO, we first introduce the concept of prediction-correction consistency as follows.

**Definition 1.** We call a variable $\boldsymbol{x}_i$ prediction-correction consistent if the predicted and reference partial solutions yield the same variable value, i.e., $\hat{\boldsymbol{x}}_i = \tilde{\boldsymbol{x}}_i$. Furthermore, we define the prediction-correction consistency of a variable as the negative of the prediction-correction discrepancy, given by $-d(p_{\boldsymbol{\theta}}(\boldsymbol{x}_i \mid \mathcal{I}), q(\boldsymbol{x}_i \mid \hat{\boldsymbol{x}}_i, \mathcal{I}))$.

We investigate the relation between UEBO and prediction-correction consistency. Our findings indicate that prediction-correction consistency serves as a useful estimator of UEBO, as demonstrated in Theorem 2, with the proof available in Appendix C.2.

**Theorem 2** (UEBO and Consistency). *Given a variable $\boldsymbol{x}_i$, UEBO is monotonically increasing with respect to the prediction-correction discrepancy. Therefore, UEBO decreases as the prediction-correction consistency increases.*

Theorem 2 illustrates that to compare the UEBOs of two variables, it suffices to compare their prediction-correction consistencies.

**(2) Variable Fixing Strategy**   We define the following consistency-based variable fixing strategy given the predicted and reference partial solutions, $\hat{\boldsymbol{x}}[P]$ and $\tilde{\boldsymbol{x}}[P]$. Specifically, we let

$$P' = \{i \in P \mid \hat{\boldsymbol{x}}_i = \tilde{\boldsymbol{x}}_i\}, \quad \boldsymbol{x}^{Corr}[P'] := \hat{\boldsymbol{x}}[P'] = \tilde{\boldsymbol{x}}[P']. \tag{7}$$

The fixing strategy outlined in Equation (7) fixes the variables that are prediction-correction consistent. We will demonstrate the significant advantages of this proposed variable fixing strategy compared to those based solely on predicted or reference solutions, emphasizing its ability to further enhance precision. We present the pseudo-code of Apollo-MILP in Algorithm 1.

**(3) Advancement of the Fixing Strategy**   Let $q(\boldsymbol{x}_i \mid \tilde{\boldsymbol{x}}_i, \hat{\boldsymbol{x}}_i, \mathcal{I})$ be the marginal distribution of the optimal solution for variable $\boldsymbol{x}_i$, given the predicted value $\hat{\boldsymbol{x}}_i$ and reference values $\tilde{\boldsymbol{x}}_i$. We outline the following consistency conditions. The condition is motivated by a classical probabilistic problem: two students provide the same answer to a multiple-choice question respectively, then the answer is more likely to be correct (see Appendix B.1 for more details). Analogous to the problem, the condition is intuitive and straightforward: we have greater confidence in the precision of the prediction $q(\boldsymbol{x}_i = 1 \mid \tilde{\boldsymbol{x}}_i = 1, \hat{\boldsymbol{x}}_i = 1, \mathcal{I})$ for the optimal variable value $\boldsymbol{x}_i^*$ when the predicted and reference values, $\hat{\boldsymbol{x}}_i$ and $\tilde{\boldsymbol{x}}_i$, yield the same result.

**Assumption 1** (Consistency Conditions). *Consistency between the predicted and reference values for variable $\boldsymbol{x}_i$ enhances the likelihood of precisely predicting the optimal solution, i.e.,*

$$q(\boldsymbol{x}_i = 1 \mid \tilde{\boldsymbol{x}}_i = 1, \hat{\boldsymbol{x}}_i = 1, \mathcal{I}) \geq q(\boldsymbol{x}_i = 1 \mid \tilde{\boldsymbol{x}}_i = 0, \hat{\boldsymbol{x}}_i = 1, \mathcal{I}), \text{ and}$$
$$q(\boldsymbol{x}_i = 1 \mid \tilde{\boldsymbol{x}}_i = 1, \hat{\boldsymbol{x}}_i = 1, \mathcal{I}) \geq q(\boldsymbol{x}_i = 1 \mid \tilde{\boldsymbol{x}}_i = 1, \hat{\boldsymbol{x}}_i = 0, \mathcal{I}). \tag{8}$$

Based on the above conditions, we analyze the effects of fixing the prediction-correction consistent variables. The proposed strategy ensures greater precision in identifying the optimal variable value.

**Theorem 3** (Precision Improvement Guarantee). *Suppose the consistency conditions (8) hold. Then, the prediction precision for variables with consistent results $q(\boldsymbol{x}_i = 1 \mid \tilde{\boldsymbol{x}}_i = 1, \hat{\boldsymbol{x}}_i = 1, \mathcal{I})$ is higher than that of variables based solely on the predicted or reference solutions, i.e.,*

$$q(\boldsymbol{x}_i = 1 \mid \tilde{\boldsymbol{x}}_i = 1, \hat{\boldsymbol{x}}_i = 1, \mathcal{I}) \geq q(\boldsymbol{x}_i = 1 \mid \tilde{\boldsymbol{x}}_i = 1, \mathcal{I}), \text{ and}$$
$$q(\boldsymbol{x}_i = 1 \mid \tilde{\boldsymbol{x}}_i = 1, \hat{\boldsymbol{x}}_i = 1, \mathcal{I}) \geq q(\boldsymbol{x}_i = 1 \mid \hat{\boldsymbol{x}}_i = 1, \mathcal{I}). \tag{9}$$

Please refer to Appendix C.3 for the proof. Finally, we examine the feasibility guarantee of Apollo-MILP. As the feasibility of PS is closely related to Problem 2, we show that our method allows for better feasibility than Problem 2, and hence the PS method.

**Corollary 4.** *Suppose we select variables to fix based on the strategy in Equation (7). If the trust-region searching problem 2 (the PS method) is feasible, then the corresponding reduced problem 6 provided by Apollo-MILP will also be feasible.*

## 5   EXPERIMENTS

In this part, we conduct extensive studies to demonstrate the effectiveness of our framework. Our method achieves significant improvements in solving performance (Section 5.2), generalization ability (Appendix H.5), and real-world applicability (Appendix H.1). Please refer to Appendix D for a detailed implementation of the methods.

---

**Algorithm 1:** Alternating Prediction-Correction Neural Solving Framework

---

**Input:** MILP Instance $\mathcal{I}$ to solve, the predictor $p_\theta$, iteration number $K$, hyperparameters
$\qquad \{(k_0^{(i)}, k_1^{(i)}, \triangle^{(i)})\}_{i=1}^K$

1   Initialize: the reduced problem $\mathcal{I}^{(0)} \leftarrow \mathcal{I}$.
2   **for** $k$ *in* $\{0, \cdots, K\}$ **do**
3      # Prediction Step
4      Obtain a predicted solution $\hat{x} \sim p_\theta(x \mid \mathcal{I}^{(k)})$ from the predictor $p_\theta$
5      Determine the partial solution $\hat{x}[P] \in \hat{x}$ to fix according to $(k_0^{(k)}, k_1^{(k)})$
6      # Correction Step
7      Construct the trust-region searching Problem 2 over $\mathcal{I}^{(k)}$ using $(k_0^{(k)}, k_1^{(k)}, \triangle^{(k)})$
8      **if** $k=K$ **then**
9         Leveraging a solver to solve the Problem 2 for the best solution $\tilde{x}^*$
10      **end**
11      **else**
12         Leveraging a solver to solve the Problem 2 for a reference solution $\tilde{x} \sim q(x \mid \hat{x}, \mathcal{I}^{(k)})$
13         Obtain $x^{Corr}[P']$ using Criterion (7)
14         Fix $x^{Corr}[P']$ in $\mathcal{I}^{(k)}$ to obtain the new reduced problem $\mathcal{I}^{(k+1)}$
15      **end**
16   **end**
17   **return** the best solution $\tilde{x}^*$

---

## 5.1   Experiment Settings

**Benchmarks**   We conduct experiments on four popular MILP benchmarks utilized in the ML4CO field: combinatorial auctions (CA) (Leyton-Brown et al., 2000), set covering (SC) (Balas & Ho, 1980), item placement (IP) (Gasse et al., 2022) and workload appointment (WA) (Gasse et al., 2022). The first two benchmarks are standard benchmarks proposed in (Gasse et al., 2019) and are commonly used to evaluate the performance of ML solvers (Gasse et al., 2019; Han et al., 2023; Huang et al., 2024). The last two benchmarks, IP and WA, come from two challenging real-world problem families used in NeurIPS ML4CO 2021 competition (Gasse et al., 2022). We use 240 training, 60 validation, and 100 testing instances, following the settings in Han et al. (2023). Please refer to Appendix F for more details on the benchmarks.

**Baselines**   We consider the following baselines in our experiments. We compare the proposed method with Neural Diving (ND) (Nair et al., 2020) and Predict-and-Search (PS) (Han et al., 2023), which we have introduced in the previous sections. Contrastive Predict-and-Search (ConPS) (Huang et al., 2024) is a strong baseline, leveraging contrastive learning to enhance the performance of PS. For ConPS, we set the ratio of positive to negative samples at ten, using low-quality solutions as negative samples. These baselines operate independently of the backbone solvers and can be integrated with traditional solvers such as SCIP (Achterberg, 2009) and Gurobi (Gurobi Optimization, 2021). Therefore, we also include SCIP and Gurobi as baselines for a comprehensive comparison. Following Han et al. (2023), Gurobi and SCIP are set to focus on finding better primal solutions.

**Metrics**   We evaluate the methods on each test instance and record the best objective value OBJ within 1,000 seconds. Following the setting in Han et al. (2023), we also run a single-thread Gurobi for 3,600 seconds and denote the best objective value as the best-known solution (BKS) to approximate the optimal value. However, we find that our method, when built on Gurobi, can identify better solutions within 1,000 seconds than Gurobi achieves in 3,600 seconds for the IP and WA benchmarks. As a result, we use the best objectives obtained by our approach as the BKS for these two benchmarks. We define the absolute primal gap as the difference between the best objective found by the solvers and the BKS, expressed as $\text{gap}_{\text{abs}} := |\text{OBJ} - \text{BKS}|$. Within the same solving time, a lower absolute primal gap indicates stronger performance.

**Implementations**   In our experiments, we conduct four rounds of iterations. The time allocated for each iteration is 100, 100, 200, and 600 seconds, respectively. We denote the size of the partial solution in the $i^{th}$ iteration by $k^{(i)} = k_0^{(i)} + k_1^{(i)}$ with $k_0^{(i)}$ variables fixed to 0 and $k_1^{(i)}$ to 1, and allow

Table 1: Comparison of solving performance between our approach and baseline methods, under a $1,000$s time limit. We build the ML approaches on Gurobi and SCIP, respectively. As we choose the challenging benchmarks with large-size instances, the solvers reach the time limit in all the experiments. We thus report the average best objective values and the absolute primal gap. '↑' indicates that higher is better, and '↓' indicates that lower is better. We mark the **best values** in bold. We also report the improvement of our method over the traditional solvers in terms of $\text{gap}_{\text{abs}}$. We find our method with a 1,000s runtime can outperform Gurobi with 3,600s runtime in IP and WA.

| | CA (BKS 97616.59) | | SC (BKS 122.95) | | IP (BKS 8.90) | | WA (BKS 704.88) | |
|---|---|---|---|---|---|---|---|---|
| | Obj ↑ | $\text{gap}_{\text{abs}}$ ↓ | Obj ↓ | $\text{gap}_{\text{abs}}$ ↓ | Obj ↓ | $\text{gap}_{\text{abs}}$ ↓ | Obj ↓ | $\text{gap}_{\text{abs}}$ ↓ |
| Gurobi | 97297.52 | 319.07 | 123.40 | 0.45 | 9.38 | 0.48 | 705.49 | 0.61 |
| ND+Gurobi | 96002.99 | 1613.59 | 123.25 | 0.29 | 9.33 | 0.43 | 705.70 | 0.82 |
| PS+Gurobi | 97358.23 | 258.36 | 123.30 | 0.35 | 9.17 | 0.27 | 705.45 | 0.57 |
| ConPS+Gurobi | 97464.10 | 152.49 | 123.20 | 0.25 | 9.09 | 0.19 | 705.37 | 0.49 |
| Ours+Gurobi | **97487.18** | **129.41** | **123.05** | **0.10** | **8.90** | **0.00** | **704.88** | **0.00** |
| Improvement | | 52.2% | | 77.8% | | 100.0% | | 100.0% |
| SCIP | 96544.10 | 1072.48 | 124.80 | 1.85 | 14.50 | 5.60 | 709.62 | 4.74 |
| ND+SCIP | 95909.50 | 1707.09 | 123.90 | 0.95 | 13.61 | 4.71 | 709.55 | 4.67 |
| PS+SCIP | 96783.62 | 832.97 | 124.35 | 1.40 | 14.25 | 5.35 | 709.39 | 4.51 |
| ConPS+SCIP | 96824.26 | 792.33 | 123.90 | 0.95 | 13.74 | 4.84 | 709.33 | 4.45 |
| Ours+SCIP | **96839.34** | **777.25** | **123.50** | **0.55** | **12.86** | **3.96** | **709.29** | **4.41** |
| Improvement | | 27.5% | | 70.2% | | 29.2% | | 6.9% |

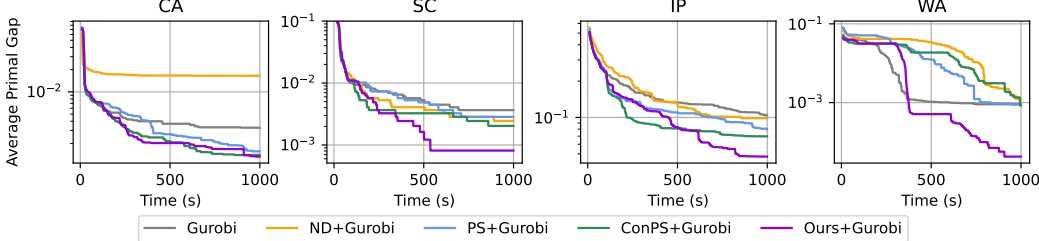

Figure 3: The primal gap of the approaches as the solving process proceeds. Our methods are implemented using Gurobi, with a time limit set to 1,000s, and we average the results across 100 testing instances. A lower primal gap for our method indicates stronger convergence performance.

$\triangle^{(i)}$ of the fixed variables to be flipped during the trust-region search. The total size of the partial solutions is given by $k_{\text{fix}} = \sum_{i=1}^{4} k^{(i)}$, which sums the partial solution sizes across all iterations. More details on hyperparameters are in Appendix G.

## 5.2 MAIN EVALUATION

**Solving Performance** To evaluate the effectiveness of the proposed method, we compare the solving performance between our framework and the baselines, under a time limit of 1,000 seconds. Table 1 presents the average best objectives found by the solvers alongside the average absolute primal gap. The instances in the IP and WA datasets possess more complex structures and larger sizes, making them more challenging for the solvers. While ND demonstrates strong performance in the CA and SC datasets, it falls short in the real-world datasets, IP and WA. ConPS serves as a robust baseline across all benchmarks, indicating that contrastive learning effectively enhances the predictor, leading to higher-quality predicted solutions. The results reveal that our proposed Apollo-MILP consistently outperforms the baselines, achieving the best objectives and the lowest gaps across the benchmarks. Specifically, Apollo-MILP reduces the absolute primal gap by over 80% compared to Gurobi and by 30% compared to SCIP. Furthermore, in the IP and WA benchmarks, our approach identifies better solutions within 1,000s than those obtained by running Gurobi for 3,600s.

**Primal Gap as a Function of Runtime** Figure 3 illustrates the curves of the average primal gap, defined as $\text{gap}_{\text{rel}} := |\text{OBJ} - \text{BKS}|/|\text{BKS}|$, throughout the solving process. Similar to the absolute

primal gap, the primal gap reflects the convergence properties of the solvers; a rapid decrease in the curves indicates superior solving performance. As shown in Figure 3, the primal gap of Apollo-MILP exhibits a gradual decrease in the early stages as it focuses on correction steps to improve the quality of partial solutions. Subsequently, the primal gap demonstrates a rapid decline, ultimately achieving the lowest gap, which highlights Appolo-MILP's strong convergence performance.

## 5.3 ABLATION STUDY

**Fixing Strategies**  To better understand Apollo-MILP, we conduct ablation studies on the variable fixing strategies. Specifically, we implement two baselines for variable fixing strategies: Direct Fixing (four rounds of direct fixing) and Multi-stage PS (four rounds of PS). We utilize the same set of hyperparameters as our method. The Multi-stage PS strategy directly fixes the variables in the predicted partial solutions $\hat{x}[P]$. The Direct Fixing strategy directly fixes the variables in the reference partial solutions $\tilde{x}[P]$. The results on the IP and WA benchmarks are presented in Table 2. The results in Table 2 show that our proposed consistency-based fixing strategy outperforms the other baselines, highlighting our method's effectiveness. Please see Appendix H.3 for more experiment results.

Table 2: Comparison of solving performance between our approach and different fixing strategies, under a $1,000$s time limit. We report the average best objective values and absolute primal gap. '↓' indicates that lower is better. We mark the **best values** in bold.

| | IP (BKS 8.90) | | WA (BKS 704.88) | |
|---|---|---|---|---|
| | Obj ↓ | $\text{gap}_{\text{abs}}$ ↓ | Obj ↓ | $\text{gap}_{\text{abs}}$ ↓ |
| Gurobi | 9.38 | 0.48 | 705.49 | 0.61 |
| PS+Gurobi | 9.17 | 0.27 | 705.45 | 0.57 |
| Direct Fixing | 9.22 | 0.32 | 705.40 | 0.52 |
| Multi-stage PS | 9.18 | 0.28 | 705.33 | 0.45 |
| Ours+Gurobi | **8.90** | **0.00** | **704.88** | **0.00** |

**Comparison with Warm-Starting Gurobi**  Warm-starting is an alternative to the trust-region search in PS and our method, in which we provide an initial feasible solution to Gurobi to guide the solving process. Gurobi can search around these start solutions or partial solutions. Warm-starting is a crucial baseline to help us understand the trust-region search. Specifically, we implement two methods, warm-starting PS (WS-PS) and warm-starting our method (WS-Ours). WS-PS passes the initial GNN prediction to Gurobi as a start solution, with hyperparameters such as $k_0$ and $k_1$ same as those we conduct in our main experiments. We also implement WS-Ours, which employs the same prediction model but replaces the trust-region search with warm-starting at each step. The results are presented in Table 19, in which we set the solving time limit as 1,000s. The results show that WS Gurobi performs comparably to PS, while WS Ours combined with Gurobi outperforms WS Gurobi, demonstrating the effectiveness of our proposed variable fixing strategy. Finally, our proposed method performs the best. The trust-region search is a more effective search method that aligns well with our framework. Please see Appendix H.3 for more experiment results.

Table 3: Comparison of solving performance between our method and the warm-starting methods, under a $1,000$s time limit. We report the average best objective values and absolute primal gap. '↓' indicates that lower is better. We mark the **best values** in bold.

| | IP (BKS 8.90) | | WA (BKS 704.88) | |
|---|---|---|---|---|
| | Obj ↓ | $\text{gap}_{\text{abs}}$ ↓ | Obj ↓ | $\text{gap}_{\text{abs}}$ ↓ |
| Gurobi | 9.38 | 0.48 | 705.49 | 0.61 |
| PS+Gurobi | 9.17 | 0.27 | 705.45 | 0.57 |
| WS-PS+Gurobi | 9.20 | 0.30 | 705.45 | 0.57 |
| WS-Ours+Gurobi | 9.13 | 0.23 | 705.40 | 0.52 |
| Ours+Gurobi | **8.90** | **0.00** | **704.88** | **0.00** |

## 6 CONCLUSION AND FUTURE WORKS

In this paper, we propose a novel ML-based solving framework (Apollo-MILP) to identify high-quality solutions for MILP problems. Apollo-MILP leverages the strengths of both Neural Diving and Predict-and-Search, alternating between prediction and correction steps to iteratively refine the predicted solutions and reduce the complexity of MILP problems. Experiments show that Apollo-MILP significantly outperforms other ML-based approaches in terms of solution quality, demonstrating strong generalization ability and promising real-world applicability.

ACKNOWLEDGMENTS

The authors would like to thank all the anonymous reviewers for their valuable suggestions. This work was supported by the National Key R&D Program of China under contract 2022ZD0119801 and the National Nature Science Foundations of China grants U23A20388 and 62021001.

ETHIC STATEMENT

This paper aims to explore the potential of an efficient MILP solving framework and obey the ICLR code of ethics. We do not foresee any direct, immediate, or negative societal impacts stemming from the outcomes of our research.

REPRODUCIBILITY STATEMENT

We provide the following information for the reproducibility of our proposed Apollo-MILP.

1. **Method.** We provide the pseudo-code of our method in Section 4.3. Moreover, we will make our source code publicly available once the paper is accepted for publication.

2. **Theoretical Proof.** We provide the proof of our theoretical results in Appendix C.

3. **Implementations.** We discuss the hyperparameters in Table 10 of Appendix G. The information on the implementation details can be found in Appendix D.

## 7 ACKNOWLEDGEMENT

The authors would like to thank all the anonymous reviewers for their insightful comments and valuable suggestions. This work was supported by the National Key R&D Program of China under contract 2022ZD0119801 and the National Nature Science Foundations of China grants U23A20388 and 62021001.

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

## A NOTATIONS

For a better understanding, we summarize and provide some key notations used in this paper in Table 4.

Table 4: Notations used in our paper.

| Notations | Descriptions |
|---|---|
| $\mathcal{I}$ | A MILP instance. |
| $\boldsymbol{x}$ | The decision variables in the MILP. |
| $\boldsymbol{x}_i$ | The $i^{th}$ component of the solution decision variable. |
| $\hat{\boldsymbol{x}}$ | The predicted solution for the MILP. |
| $\tilde{\boldsymbol{x}}$ | The reference solution for the MILP. |
| $\boldsymbol{x}[P]$ | The partial solution the same variable values as $\boldsymbol{x}$ in the index set $P$. |
| $\boldsymbol{x}^{Corr}[P]$ | The corrected partial solution given the index set $P$. |
| $q(\boldsymbol{x} \mid \mathcal{I})$ | The distribution of the optimal solution given a MILP instance $\mathcal{I}$. |
| $p_\theta(\boldsymbol{x} \mid \mathcal{I})$ | The predicted marginal probabilities for the solution, given an instance $\mathcal{I}$. |
| $q(\boldsymbol{x} \mid \hat{\boldsymbol{x}}, \mathcal{I})$ | The distribution of the reference solution given instance $\mathcal{I}$ and predicted solution $\hat{\boldsymbol{x}}$. |
| $D_{KL}(\cdot\|\cdot)$ | The KL divergence of two distributions. |
| $\mathcal{H}(\cdot)(\cdot)$ | The entropy of a distribution. |
| $d(\cdot, \cdot)$ | The cross-entropy of two distributions. |

## B THE SKETCH OF DERIVATION OF THE VARIABLE FIXING STRATEGY

### B.1 THE MOTIVATION OF ASSUMPTION 1: FROM A CLASSICAL PROBLEM

In this part, We will provide some evidence to support Assumption 1 from both an intuitive perspective for better understanding.

**Example 1.** Consider two students, A and B, participating in a math competition with a multiple-choice question offering two options. Let the probability that student A answers correctly be $p$, and the probability for student B be $q$. Since both have prepared for the exam, we assume $p, q > 0.5$. If they provide the same answer, what is the probability that their answer is correct? Conversely, if they provide different answers, what is the probability that student A is correct?

**Answer.** When both students provide the same answer, the probability that both are correct is given by:

$$\mathbb{P}(\text{A,B are correct} \mid \text{A,B provide the same answers})$$
$$= \frac{\mathbb{P}(\text{A,B are correct and provide the same answers})}{\mathbb{P}(\text{A,B provide the same answers})} = \frac{pq}{pq + (1-p)(1-q)}. \tag{10}$$

Given that their answers are different, the probability that A is correct is

$$\mathbb{P}(\text{A is correct} \mid \text{A,B provide different answers})$$
$$= \frac{\mathbb{P}(\text{A is correct, and A,B provide different answers})}{\mathbb{P}(\text{A,B provide different answers})} = \frac{p(1-q)}{p(1-q) + (1-p)q}. \tag{11}$$

A simple calculation reveals that

$$\mathbb{P}(\text{A,B are correct} \mid \text{A,B provide the same answers}) > \mathbb{P}(\text{A is correct} \mid \text{A,B provide different answers}), \tag{12}$$

suggesting that if both students provide the same answer, it is more likely to be correct.

Returning to Assumption 1, we can draw an analogy: the predicted solution and the reference solution correspond to the answers given by the two students, while the optimal solution represents the correct answer. For simplicity, we treat these as independent. As shown in the example, ifs

$p, q > 0.5$, then a consistent answer yields higher precision than differing answers. Given that the predictor is well-trained, we believe that the precision will exceed 0.5. Similarly, the precision of the traditional solver is also expected to be greater than 0.5.

$$q(x_i = 1 \mid \tilde{x}_i = 1, \hat{x}_i = 1, \mathcal{I}) \geq q(x_i = 1 \mid \tilde{x}_i = 0, \hat{x}_i = 1, \mathcal{I}), \text{ and}$$
$$q(x_i = 1 \mid \tilde{x}_i = 1, \hat{x}_i = 1, \mathcal{I}) \geq q(x_i = 1 \mid \tilde{x}_i = 1, \hat{x}_i = 0, \mathcal{I}). \tag{13}$$

### B.2 THE SKETCH OF DERIVATION

In this paper, we propose the UEBO metric to determine the variables to fix. In practice, we do not calculate UEBO directly. Instead, we propose to approximate UEBO in Section 4.3.

- First, we introduce a concept called **prediction-correction consistency** (Definition 1 in Section 4.3).

$$-d(p_\theta(\boldsymbol{x}_i \mid \mathcal{I}), q(\boldsymbol{x}_i \mid \hat{\boldsymbol{x}}_i, \mathcal{I})) = q(\boldsymbol{x}_i = 0 \mid \hat{\boldsymbol{x}}_i, \mathcal{I}) \log(1 - p_\theta(\boldsymbol{x}_i = 1 \mid \mathcal{I}))$$
$$+ q(\boldsymbol{x}_i = 1 \mid \hat{\boldsymbol{x}}_i, \mathcal{I}) \log(p_\theta(\boldsymbol{x}_i = 1 \mid \mathcal{I})), \tag{14}$$

  which is the negative of cross entropy loss using reference solutions as 'labels'.

- Second, we also show that **UEBO decreases as the prediction-correction consistency increases** (Theorem 2 in Section 4.3). Using this property, to compare the UEBO of two variables, we just need to **compare their prediction-correction consistency**, with higher prediction-correction consistency indicating a lower UEBO. Here we provide a simple numerical example to explain this process.

**Example 2.** We suppose $k_0 = 0$, $k_1 = 3$ and $\triangle = 1$ for simplicity. GNN prediction for an instance with five binary variables $\boldsymbol{x}_i$ ($i = 1, \ldots, 5$) is $[0.9, 0.8, 0.7, 0.6, 0.5]$. First, we add the following constraints to the instance,

$$\boldsymbol{x}_1 - 1 \leq \alpha_1, 1 - \boldsymbol{x}_1 \leq \alpha_1,$$
$$\boldsymbol{x}_2 - 1 \leq \alpha_2, 1 - \boldsymbol{x}_2 \leq \alpha_2,$$
$$\boldsymbol{x}_3 - 1 \leq \alpha_3, 1 - \boldsymbol{x}_3 \leq \alpha_3,$$
$$\alpha_1 + \alpha_2 + \alpha_3 \leq 1. \tag{15}$$

Suppose that the reference solution is $[\tilde{\boldsymbol{x}}_1, \tilde{\boldsymbol{x}}_2, \tilde{\boldsymbol{x}}_3, \tilde{\boldsymbol{x}}_4, \tilde{\boldsymbol{x}}_5] = [1, 0, 1, 1, 1]$. Thus, we have the prediction-correction consistency

$$-d(p_\theta(\boldsymbol{x}_1 \mid \mathcal{I}), q(\boldsymbol{x}_1 \mid \hat{\boldsymbol{x}}_1, \mathcal{I})) = 1 \times \log(0.9) + 0 \times \log(0.1) = \log(0.9),$$
$$-d(p_\theta(\boldsymbol{x}_2 \mid \mathcal{I}), q(\boldsymbol{x}_2 \mid \hat{\boldsymbol{x}}_2, \mathcal{I})) = 0 \times \log(0.8) + 1 \times \log(0.2) = \log(0.2), \tag{16}$$
$$-d(p_\theta(\boldsymbol{x}_3 \mid \mathcal{I}), q(\boldsymbol{x}_3 \mid \hat{\boldsymbol{x}}_3, \mathcal{I})) = 1 \times \log(0.7) + 0 \times \log(0.3) = \log(0.7).$$

Thus, the variables $\boldsymbol{x}_1$ and $\boldsymbol{x}_3$ have higher prediction-correction consistencies and thus lower UEBO. We have more confidence to fix $\boldsymbol{x}_1$ and $\boldsymbol{x}_3$.

- Third, we step further to **simplify the fixing rule**. Intuitively, a well-trained model should satisfy the property that if the predicted partial solution $\hat{\boldsymbol{x}}_i = 1$, then $p_\theta(\boldsymbol{x}_i \mid \mathcal{I})$ should be larger than 0.5; if the predicted partial solution $\hat{\boldsymbol{x}}_i = 0$, then $p_\theta(\boldsymbol{x}_i \mid \mathcal{I})$ should be smaller than 0.5. Using this observation, we find the inequality

$$-d(p_\theta(\boldsymbol{x}_i \mid \mathcal{I}), q(\tilde{\boldsymbol{x}}_i \mid \hat{\boldsymbol{x}}_i, \mathcal{I})) \geq -d(p_\theta(\boldsymbol{x}_j \mid \mathcal{I}), q(\tilde{\boldsymbol{x}}_j \mid \hat{\boldsymbol{x}}_j, \mathcal{I})) \tag{17}$$

always hold given $\hat{\boldsymbol{x}}_i = \tilde{\boldsymbol{x}}_i$, $\hat{\boldsymbol{x}}_j \neq \tilde{\boldsymbol{x}}_j$ with $\hat{\boldsymbol{x}}_i = \hat{\boldsymbol{x}}_j$. To see this, assume that $\hat{\boldsymbol{x}}_i = \hat{\boldsymbol{x}}_j = 1$, we have $p_\theta(\boldsymbol{x}_i \mid \mathcal{I}) \geq 0.5$ and $p_\theta(\boldsymbol{x}_j \mid \mathcal{I}) \geq 0.5$. Then, we have

$$-d(p_\theta(\boldsymbol{x}_i \mid \mathcal{I}), q(\tilde{\boldsymbol{x}}_i \mid \hat{\boldsymbol{x}}_i, \mathcal{I})) = \log(p_\theta(\boldsymbol{x}_i \mid \mathcal{I})) \geq \log(0.5)$$
$$\geq \log(1 - p_\theta(\boldsymbol{x}_j \mid \mathcal{I})) = -d(p_\theta(\boldsymbol{x}_j \mid \mathcal{I}), q(\tilde{\boldsymbol{x}}_j \mid \hat{\boldsymbol{x}}_j, \mathcal{I})). \tag{18}$$

Thus, we propose the fixing strategy in Equation (7). The proposed fixing strategy fixes variables satisfying $\hat{\boldsymbol{x}}_i = \tilde{\boldsymbol{x}}_i$, which are indeed the variables with **high prediction-correction consistency** and thus **low UEBO**.

**Example 3.** In Example 2, $\hat{\boldsymbol{x}}_1 = \tilde{\boldsymbol{x}}_1 = 1$, $\hat{\boldsymbol{x}}_2 \neq \tilde{\boldsymbol{x}}_2$, $\hat{\boldsymbol{x}}_3 = \tilde{\boldsymbol{x}}_3 = 1$. Thus, we fix $\boldsymbol{x}_1$ and $\boldsymbol{x}_3$ to value 1. This result coincides with that given in Example 2.

# C PROOF OF PROPOSITIONS AND THEOREMS.

## C.1 PROOF OF PROPOSITION 1

We show the results by direct calculations,

$$
\begin{aligned}
D_{\mathrm{KL}}\left(\ p_{\boldsymbol{\theta}}(\boldsymbol{x}_i \mid \mathcal{I}) \| q(\boldsymbol{x}_i \mid \mathcal{I})\right) &= \int_{\mathcal{I}} \int_{\boldsymbol{x}_i} p_{\boldsymbol{\theta}}(\boldsymbol{x}_i \mid \mathcal{I}) p(\mathcal{I}) \log\left(\frac{p_{\boldsymbol{\theta}}(\boldsymbol{x}_i \mid \mathcal{I})}{q(\boldsymbol{x}_i \mid \mathcal{I})}\right) d\boldsymbol{x}_i d\mathcal{I} \\
&= \int_{\mathcal{I}} \int_{\boldsymbol{x}_i} p_{\boldsymbol{\theta}}(\boldsymbol{x}_i \mid \mathcal{I}) p(\mathcal{I}) \log(p_{\boldsymbol{\theta}}(\boldsymbol{x}_i \mid \mathcal{I})) d\boldsymbol{x}_i d\mathcal{I} - \int_{\mathcal{I}} \int_{\boldsymbol{x}_i} p_{\boldsymbol{\theta}}(\boldsymbol{x}_i \mid \mathcal{I}) p(\mathcal{I}) \log(q(\boldsymbol{x}_i \mid \mathcal{I})) d\boldsymbol{x}_i d\mathcal{I} \\
&= -\mathcal{H}(p_{\boldsymbol{\theta}}(\boldsymbol{x}_i \mid \mathcal{I})) - \int_{\mathcal{I}} \int_{\boldsymbol{x}_i} p_{\boldsymbol{\theta}}(\boldsymbol{x}_i \mid \mathcal{I}) p(\mathcal{I}) \log\left(\int_{\hat{\boldsymbol{x}}_i} q(\boldsymbol{x}_i \mid \hat{\boldsymbol{x}}_i, \mathcal{I}) p_{\boldsymbol{\theta}}(\hat{\boldsymbol{x}}_i \mid \mathcal{I}) d\hat{\boldsymbol{x}}_i\right) d\boldsymbol{x}_i d\mathcal{I} \\
&\leq -\mathcal{H}(p_{\boldsymbol{\theta}}(\boldsymbol{x}_i \mid \mathcal{I})) - \int_{\mathcal{I}} \int_{\boldsymbol{x}_i} \int_{\hat{\boldsymbol{x}}_i} p_{\boldsymbol{\theta}}(\boldsymbol{x}_i \mid \mathcal{I}) p(\mathcal{I}) p_{\boldsymbol{\theta}}(\hat{\boldsymbol{x}}_i \mid \mathcal{I}) \log(q(\boldsymbol{x}_i \mid \hat{\boldsymbol{x}}_i, \mathcal{I})) d\hat{\boldsymbol{x}}_i d\boldsymbol{x}_i d\mathcal{I} \\
&= -\mathcal{H}(p_{\boldsymbol{\theta}}(\boldsymbol{x}_i \mid \mathcal{I})) + d(p_{\boldsymbol{\theta}}(\boldsymbol{x}_i \mid \mathcal{I}), q(\boldsymbol{x}_i \mid \hat{\boldsymbol{x}}_i, \mathcal{I})).
\end{aligned}
\tag{19}
$$

## C.2 PROOF OF THEOREM 2

We analyze the monotony property of UEBO as follows. First, we suppose that the variable $\tilde{\boldsymbol{x}}_i$ takes value 1 in the reference solution, i.e., $\tilde{\boldsymbol{x}}_i = 1$. Thus, UEBO is in the form of

$$
\begin{aligned}
\mathrm{UEBO}(p_{\theta}(\hat{\boldsymbol{x}}_i \mid \mathcal{I}), q(\tilde{\boldsymbol{x}}_i \mid \hat{\boldsymbol{x}}_i, \mathcal{I}))\Big|_{\tilde{\boldsymbol{x}}_i = 1} &= \mathcal{H}(p_{\boldsymbol{\theta}}(\hat{\boldsymbol{x}}_i \mid \mathcal{I})) + d(p_{\boldsymbol{\theta}}(\hat{\boldsymbol{x}}_i \mid \mathcal{I}), q(\tilde{\boldsymbol{x}}_i \mid \hat{\boldsymbol{x}}_i, \mathcal{I})) \\
&= -p_{\boldsymbol{\theta}}(\hat{\boldsymbol{x}}_i \mid \mathcal{I}) \log p_{\boldsymbol{\theta}}(\hat{\boldsymbol{x}}_i \mid \mathcal{I}) - (1 - p_{\boldsymbol{\theta}}(\hat{\boldsymbol{x}}_i \mid \mathcal{I})) \log(1 - p_{\boldsymbol{\theta}}(\hat{\boldsymbol{x}}_i \mid \mathcal{I})) - \log p_{\boldsymbol{\theta}}(\hat{\boldsymbol{x}}_i \mid \mathcal{I}) \\
&= -(1 + p_{\boldsymbol{\theta}}(\hat{\boldsymbol{x}}_i \mid \mathcal{I})) \log p_{\boldsymbol{\theta}}(\hat{\boldsymbol{x}}_i \mid \mathcal{I}) - (1 - p_{\boldsymbol{\theta}}(\hat{\boldsymbol{x}}_i \mid \mathcal{I})) \log(1 - p_{\boldsymbol{\theta}}(\hat{\boldsymbol{x}}_i \mid \mathcal{I})).
\end{aligned}
\tag{20}
$$

Differentiating UEBO with respect to the predicted logit $p_{\theta}(\hat{\boldsymbol{x}}_i \mid \mathcal{I})$, we have

$$
\frac{d\mathrm{UEBO}(p_{\theta}(\hat{\boldsymbol{x}}_i \mid \mathcal{I})\Big|_{\tilde{\boldsymbol{x}}_i = 1}}{dp_{\theta}(\hat{\boldsymbol{x}}_i \mid \mathcal{I})} = \log\left(\frac{1}{p_{\theta}(\hat{\boldsymbol{x}}_i \mid \mathcal{I})} - 1\right) - \frac{1}{p_{\theta}(\hat{\boldsymbol{x}}_i \mid \mathcal{I})} \leq 0,
\tag{21}
$$

where $p_{\theta}(\hat{\boldsymbol{x}}_i \mid \mathcal{I})$ takes values between $[0, 1]$. Therefore, UEBO decreases as $p_{\theta}(\hat{\boldsymbol{x}}_i \mid \mathcal{I})$ becomes larger in $[0, 1]$. As $p_{\theta}(\hat{\boldsymbol{x}}_i \mid \mathcal{I})$ grows, the cross-entropy term $d(p_{\theta}(\hat{\boldsymbol{x}}_i \mid \mathcal{I}), q(\tilde{\boldsymbol{x}}_i \mid \hat{\boldsymbol{x}}_i, \mathcal{I})) = -\log p_{\boldsymbol{\theta}}(\hat{\boldsymbol{x}}_i \mid \mathcal{I})$ decreases and the prediction-correction consistency becomes higher.

Similarly, the proof of case that the variable $\tilde{\boldsymbol{x}}_i$ takes value 0 in the reference solution follows the same step, and we thus show that UEBO has a negative correlation with the prediction-correction consistency.

## C.3 PROOF OF THEOREM 3

We first expand the right-hand side of the inequalities.

$$
\begin{aligned}
&q(\boldsymbol{x}_i^* = 1 \mid \tilde{\boldsymbol{x}}_i = 1, \mathcal{I}) \\
=&q(\boldsymbol{x}_i^* = 1 \mid \tilde{\boldsymbol{x}}_i = 1, \hat{\boldsymbol{x}}_i = 1, \mathcal{I}) q(\hat{\boldsymbol{x}}_i = 1 \mid \tilde{\boldsymbol{x}}_i = 1, \mathcal{I}) \\
&+ q(\boldsymbol{x}_i^* = 1 \mid \tilde{\boldsymbol{x}}_i = 1, \hat{\boldsymbol{x}}_i = 0, \mathcal{I}) q(\hat{\boldsymbol{x}}_i = 0 \mid \tilde{\boldsymbol{x}}_i = 1, \mathcal{I}) \\
\leq&q(\boldsymbol{x}_i^* = 1 \mid \tilde{\boldsymbol{x}}_i = 1, \hat{\boldsymbol{x}}_i = 1, \mathcal{I}) q(\hat{\boldsymbol{x}}_i = 1 \mid \tilde{\boldsymbol{x}}_i = 1, \mathcal{I}) \\
&+ q(\boldsymbol{x}_i^* = 1 \mid \tilde{\boldsymbol{x}}_i = 1, \hat{\boldsymbol{x}}_i = 1, \mathcal{I}) q(\hat{\boldsymbol{x}}_i = 0 \mid \tilde{\boldsymbol{x}}_i = 1, \mathcal{I}),
\end{aligned}
\tag{22}
$$

where the inequality holds by the consistency conditions (8). Thus we have

$$
\begin{aligned}
&q(\boldsymbol{x}_i^* = 1 \mid \tilde{\boldsymbol{x}}_i = 1, \mathcal{I}) \\
\leq&q(\boldsymbol{x}_i^* = 1 \mid \tilde{\boldsymbol{x}}_i = 1, \hat{\boldsymbol{x}}_i = 1, \mathcal{I}) \left(q(\hat{\boldsymbol{x}}_i = 1 \mid \tilde{\boldsymbol{x}}_i = 1, \mathcal{I}) + q(\hat{\boldsymbol{x}}_i = 0 \mid \tilde{\boldsymbol{x}}_i = 1, \mathcal{I})\right) \\
=&q(\boldsymbol{x}_i^* = 1 \mid \tilde{\boldsymbol{x}}_i = 1, \hat{\boldsymbol{x}}_i = 1, \mathcal{I}).
\end{aligned}
\tag{23}
$$

## C.4 Proof of Feasibility Guarantee

Since the trust-region searching problem 2 is feasible, we can obtain a feasible solution as a reference solution, denoted by $\tilde{x}$. Thus, $\tilde{x}$ satisfies the constraint in Problem 2, i.e.,

$$A\tilde{x} \leq b, \quad l \leq \tilde{x} \leq u, \quad \tilde{x}[P] \in \mathcal{B}_P(\hat{x}[P], \triangle), \quad \tilde{x} \in \mathbb{Z}^p \times \mathbb{R}^{n-p}.$$

According to the consistency-based variable fixing strategy, we fix the variable $x[P']$ in the instance $\mathcal{I}$ to values $\tilde{x}[P']$, where $P'$ is a subset of $P$ satisfying $\tilde{x}[P'] = \hat{x}[P']$. Therefore, we have

$$A\tilde{x} \leq b, \quad l \leq \tilde{x} \leq u, \quad \tilde{x}[P'] = x^{Corr}[P'], \quad \tilde{x} \in \mathbb{Z}^p \times \mathbb{R}^{n-p}.$$

This implies that $\tilde{x}$ is also a feasible solution of Problem 6.

## D Implementation of Our Methods and the Baselines

Machine learning has made great progress in a broad range of areas (Mao et al., 2025a;b; Dong et al., 2025; Bai et al., 2025; Wang et al., 2025; 2024d;c;e; 2023c; 2022; Yang et al., 2022; Liu et al., 2023b; Wang et al., 2023a; Yang et al., 2024) and graph neural networks have shown great performance in solving MILPs. The PS models used in this paper align with those outlined in the original papers Han et al. (2023). We use the code in `https://github.com/sribdcn/Predict-and-Search` MILP method to implement PS. For the PS predictor, we leverage a graph neural network (Kipf & Welling, 2017; Shi et al., 2023; Sui et al., 2024; Mao et al., 2024; Shi et al., 2025; 2024) comprising four half-convolution layers. The codes of the original work of ND (Nair et al., 2020) and ConPS (Huang et al., 2024) are not publicly available. We try our best to reproduce these baselines and tune the hyperparameters for testing. We conducted all the experiments on a single machine with NVidia GeForce GTX 3090 GPUs and Intel(R) Xeon(R) E5-2667 V4CPUs 3.20GHz.

In the training process of the predictors, we set the initial learning rate to be 0.001 and the training epoch to be 10,000 with early stopping. To collect the training data, we run a single thread Gurobi on each training and validation instance to for 3,600 seconds and record the best 50 solutions.

The partial solution size parameter $(k_0, k_1)$ and neighborhood parameter $\Delta$ are two important parameters in PS. The partial solution size parameter $(k_0, k_1, \Delta)$ represents the numbers of variables fixed with values 0 and 1 in a partial solution. The neighborhood parameter $\Delta$ defines the radius of the searching neighborhood. We list these two parameters used in our experiments in Table 5.

Table 5: The partial solution size parameter $(k_0, k_1)$ and neighborhood parameter $\Delta$.

| Benchmark | CA | SC | IP | WA |
|---|---|---|---|---|
| PS+Gurobi | (600,0,1) | (2000,0,100) | (400,5,10) | (0,500,10) |
| ConPS+Gurobi | (900,0,50) | (1000,0,200) | (400,5,3) | (0,500,10) |
| PS+SCIP | (400,0,10) | (2000,0,100) | (400,5,1) | (0,600,5) |
| ConPS+SCIP | (900,0,50) | (1000,0,200) | (400,5,3) | (0,400,50) |

For our Apollo-MILP, the training process follows a similar approach, but we incorporate data augmentation to align with the testing distributions. For each original training instance with $n$ variables, we select the best solution $x^*$ from the solution pool $\mathcal{S}$. Then, we randomly sample a fixing ratio $\alpha \in [0.3, 0.7]$ and an index set $P_\alpha$ of variables, where the index set contains $\alpha n$ elements. Finally, we enforce the variables in the set $\mathcal{V}_\alpha$ to the corresponding values in $x^*$, resulting in $x[P_\alpha] = x^*[P_\alpha]$. By varying the ratio $\alpha$, we generate five reduced augmented instances from each training instance.

## E Details on Bipartite Graph Representations

The bipartite instance graph representation utilized in this paper closely aligns with that presented in the PS paper Han et al. (2023). We list the graph features in Table 6.

Table 6: The variable features, constraint features, and edge features used for the predictor.

| Index | Variable Feature Name | Description |
|---|---|---|
| 0 | Objective | Normalized objective coefficient |
| 1 | Variable coefficient | Average variable coefficient in all constraints |
| 2 | Variable degree | Degree of the variable node in the bipartite graph representation |
| 3 | Maximum variable coefficient | Maximum variable coefficient in all constraints |
| 4 | Minimum variable coefficient | Minimum variable coefficient in all constraints |
| 5 | Variable type | Whether the variable is an integer variable or not) |
| 6-17 | Position embedding | Binary encoding of the order of appearance for each variable among all variables. |

| Index | Constraint Feature Name | Description |
|---|---|---|
| 0 | Constraint coefficient | Average of all coefficients in the constraint |
| 1 | Constraint degree | Degree of constraint nodes |
| 2 | Bias | Normalized right-hand-side of the constraint |
| 3 | Sense | The sense of the constraint |

| Index | Constraint Feature Name | Description |
|---|---|---|
| 0 | Coefficient | Constraint coefficient |

# F  DETAILS ON THE BENCHMARKS

## F.1  BENCHMARKS IN MAIN EVALUATION

The CA and SC benchmark instances are generated following the process described in Gasse et al. (2019). Specifically, the CA instances were generated using the algorithm from Leyton-Brown et al. (2000), and the SC instances were generated using the algorithm presented in Balas & Ho (1980). The IP and WA instances are obtained from the NeurIPS ML4CO 2021 competition (Gasse et al., 2022). The statistical information for all the instances is provided in Table 7.

Table 7: Statistical information of the benchmarks we used in this paper.

| | CA | SC | IP | WA |
|---|---|---|---|---|
| Constraint Number | 2593 | 3000 | 195 | 64306 |
| Variable Number | 1500 | 5000 | 1083 | 61000 |
| Number of Binary Variables | 1500 | 5000 | 1050 | 1000 |
| Number of Continuous Variables | 0 | 0 | 33 | 60000 |
| Number of Integer Variables | 0 | 0 | 0 | 0 |

## F.2  BENCHMARKS IN USED FOR GENERALIZATION

We generate larger CA and SC instances to evaluate the generalization ability of the approaches. We use the code in Gasse et al. (2019) for data generation. Specifically, the generated CA instances have an average of 2,596 constraints and 4,000 variables, and the SC instances have 6,000 constraints and 10,000 variables. These instances are considerably larger than the training instances.

## F.3  SUBSET OF MIPLIB

We construct a subset of MIPLIB (Gleixner et al., 2021) to evaluate the solvers' ability to handle challenging real-world instances. Specifically, we select instances based on their similarity, which is measured by 100 human-designed features (Gleixner et al., 2021). Instances with presolving times exceeding 300 seconds or those that exceed GPU memory limits during the inference process are discarded. Inspired by the IIS dataset used in Wang et al. (2024a), we develop a refined IIS dataset

containing eleven instances. We divide this dataset into training and testing sets, comprising eight training instances and three testing instances (ramos3, scpj4scip, and scpl4). Detailed information on the IIS dataset can be found in Table 8.

Table 8: Statistical information of the instances in the constructed IIS dataset.

| Instance Name | Constraint Number | Variable Number | Nonzero Coefficient Number |
|---|---|---|---|
| ex1010-pi | 1468 | 25200 | 102114 |
| fast0507 | 507 | 63009 | 409349 |
| glass-sc | 6119 | 214 | 63918 |
| iis-glass-cov | 5375 | 214 | 56133 |
| iis-hc-cov | 9727 | 297 | 142971 |
| ramos3 | 2187 | 2187 | 32805 |
| scpj4scip | 1000 | 99947 | 999893 |
| scpk4 | 2000 | 100000 | 1000000 |
| scpl4 | 2000 | 200000 | 2000000 |
| seymour | 4944 | 1372 | 33549 |
| v150d30-2hopcds | 7822 | 150 | 103991 |

## F.4 MORE BENCHMARKS

To demonstrate the effectiveness of our method, we include three more benchmarks. The statistical information is in Table 9.

**SCUC dataset** This dataset comes from the Energy Electronics Industry Innovation Competition. The benchmark contains large-scale instances from real-world power systems.

**Smaller-size CA dataset** This dataset is the CA dataset with the same sizes as the 'Hard CA' in Gasse et al. (2019). This dataset has a lower instance size and computational hardness than the CA dataset used in our paper. All the methods can solve the problems within the time limit, and we thus report the solving time.

**APS dataset** This dataset is from an anonymous commercial enterprise containing real-world production scheduling problems in the factory. The instances in APS are general integer programming problems, containing binary, continuous, and general integer variables.

Table 9: Statistical information of the benchmarks.

| | SCUC | Small-size CA | APS |
|---|---|---|---|
| Constraint Number | 27835 | 576 | 31296 |
| Variable Number | 19807 | 1500 | 31344 |
| Number of Binary Variables | 9295 | 1500 | 1500 |
| Number of Continuous Variables | 10512 | 0 | 15984 |
| Number of Integer Variables | 0 | 0 | 9600 |

## G HYPERPARAMETERS

We report the hyperparameters $(k_0^{(i)}, k_1^{(i)}, \triangle^{(i)})$ of Apollo-MILP used in Table 10.

Table 10: Hyperparameters $(k_0^{(i)}, k_1^{(i)}, \triangle^{(i)})$ for Apollo-MILP.

|  | CA | SC | IP | WA |
|---|---|---|---|---|
| Iteration 1 | (400,0,60) | (1000,0,200) | (100,20,50) | (20,200,100) |
| Iteration 2 | (200,0,30) | (500,0,100) | (40,15,20) | (10,100,50) |
| Iteration 3 | (100,0,15) | (250,0,50) | (20,15,10) | (10,5,5) |
| Iteration 4 | (50,0,10) | (10,0,5) | (5,50,30) | (1,10,5) |

## H ADDITIONAL EXPERIMENT RESULTS

### H.1 REAL-WORLD DATASET

To further demonstrate the applicability of Apollo-MILP, we conduct experiments on instances from MIPLIB (Gleixner et al., 2021), a challenging real-world dataset. Due to the heterogeneous nature of the instances in MIPLIB, applying ML-based solvers directly to the entire dataset can be difficult. However, we can focus on a subset of MIPLIB that contains similar instances (Wang et al., 2023b; 2024a). For more information on the selected MILP subset, referred to as IIS, please see Appendix F.3. This subset consists of eleven challenging real-world instances. Notice that we need to carefully tune the hyperparameters $(k_0, k_1, \triangle)$ for the ML baselines, as improper hyperparameters can easily lead to infeasibility. While Appolo-MILP exhibits a strong adaptation across different hyperparameters. We report the solving performance of the solvers in Table 11, where Apollo-MILP significantly outperforms other baselines, showcasing its promising potential for real-world applications. We also report the detailed results of the real-world challenging MIPLIB dataset mentioned. We set the time limit to 3,600 seconds and ran two iterations with 1,000 and 2,600 seconds. Given the variation in instance sizes within the dataset, we set $\triangle = 1000$ and the proportion of fixed variables to $(\alpha_0, \alpha_1) = (0.8, 0)$, which means that we fix 0.8 of the binary variables to 0 in the first round.

Table 11: The results in the IIS dataset, which is used in Wang et al. (2024a) and is a subset of MIPLIB. We build the ML approaches on Gurobi and set the solving time limit to 3,600s.

|  | Obj $\downarrow$ | gap$_{abs}$ $\downarrow$ |
|---|---|---|
| Gurobi | 214.00 | 23.00 |
| ND | 213.00 | 22.00 |
| PS | 211.00 | 20.00 |
| ConPS | 211.00 | 20.00 |
| Ours | **209.33** | **18.33** |

Table 12: The best objectives found by the approaches on each test instance in IIS. *BKS* represents the best objectives from the website of MIPLIB https://miplib.zib.de/index.html.

|  | BKS | Gurobi | ND | PS | ConPS | Ours |
|---|---|---|---|---|---|---|
| ramos3 | 186.00 | 233.00 | 233.00 | 225.00 | 225.00 | **224.00** |
| scpj4scip | 128.00 | 132.00 | **131.00** | 133.00 | 133.00 | **131.00** |
| scpl4 | 259.00 | 277.00 | 275.00 | 275.00 | 275.00 | **273.00** |

### H.2 HYPERPARAMETER ANALYSIS

We investigate the impact of hyperparameters in our proposed framework. In this part, we conduct extensive experiments to analyze the impact of hyperparameters, including the partial solution size, neighborhood parameter, iteration time, iteration round number, and the use of data augmentation. For all experiments, we implement the method using Gurobi and set a time limit of 1,000 seconds.

**Partial Solution Size Parameters** We examine the effects of the partial solution size parameter $k_{\text{fix}}$ in the CA benchmark. As noted in Huang et al. (2024), fixing $k_1^{(i)} = 0$ always yields better

solutions. Therefore, we focus on the effects of $k_0^{(i)}$ while keeping $k_1^{(i)} = 0$ and $\triangle = (60, 30, 15, 0)$ for four rounds of iterations. Our findings, presented in Table 13, indicate that a fixing coverage of 50% of variables yields the best performance. This optimal coverage balances the risk of the solution prediction methods becoming trapped in low-quality neighborhoods——common with high coverage of fixed variables——while avoiding ineffective problem reduction associated with low coverage.

Table 13: The solving performance with different partial solution size parameters $k_{\text{fix}}$ on the CA benchmark, under the time limit of 1,000 seconds. The coverage rate implies the approximate proportion of fixing variables.

|  | Obj ↑ | $\text{gap}_{\text{abs}}$ ↓ |
| --- | --- | --- |
| Coverage 85% | 96950.55 | 666.04 |
| Coverage 70% | 96929.98 | 686.61 |
| Coverage 50% | **97487.18** | **129.41** |
| Coverage 30% | 97359.09 | 257.50 |

**Neighborhood Parameter**    We examine how the choice of neighborhood parameters affects solving performance. From Table 14, we can see that when $\triangle$ is small, increasing its value can enhance performance since searching within a larger area may yield higher-quality solutions. However, when $\triangle$ is too large, performance decreases, as the expanded trust region results in a larger search space, leading to inefficiencies in the search process.

Table 14: The effects of neighborhood parameters on the solving performance.

|  | Obj ↑ | $\text{gap}_{\text{abs}}$ ↓ |
| --- | --- | --- |
| 60% of Fixing Numbers | 97297.52 | 319.07 |
| 50% of Fixing Numbers | 97343.47 | 273.12 |
| 20% of Fixing Numbers | **97487.18** | **129.41** |
| 10% of Fixing Numbers | 97019.17 | 597.12 |

**Iteration Time**    We investigate the relationship between iteration time and performance in Table 15, setting a time limit of 1,000 seconds. The early iterations focus on identifying a high-quality feasible solution to reduce the search space, while the final iteration aims to exploit the optimal solution within this reduced space. Through four iterations, we find that the last iteration, allocated 600 seconds, yields the best performance. As the exploitation time increases, the algorithm is more likely to search the reduced space thoroughly. However, extending the exploitation time reduces the available time to enhance the predicted solution during the early stages, which can lead to a decline in the quality of the predicted solutions.

Table 15: The effects of iteration time on the solving performance.

|  | Obj ↑ | $\text{gap}_{\text{abs}}$ ↓ |
| --- | --- | --- |
| (25,25,50,900) | 96741.04 | 875.55 |
| (50,50,100,800) | 96889.70 | 726.89 |
| (100,100,200,600) | **97487.18** | **129.41** |
| (150,150,300,400) | 97353.35 | 263.24 |

**The Rounds of Iterations**    We conduct experiments on the rounds of iterations. We fix the solving time limit to 1,000s and compare different rounds of iterations in the CA dataset. Given an iteration round, we set the same search time across the iterations. The results are presented in Table 16, which indicates the four rounds of iterations have the best performance.

**Data Augmentation**    The distribution of the reduced problems in each iteration may differ from that of the original problems. As pointed out in Section 4.1, we employ data augmentation to align the distributional shifts. We conduct experiments to evaluate the effect of data augmentation in

Table 16: The effects of rounds of the iterations on solving performance.

|  | Obj ↑ | $\text{gap}_{abs}$ ↓ |
|---|---|---|
| (500,500) | 97132.39 | 484.20 |
| (333.3,333.3,333.3) | 97349.74 | 266.85 |
| (250,250,250,250) | **97388.21** | **228.38** |
| (200,200,200,200,200) | 96889.70 | 726.89 |

Table 17. We use the CA dataset and set the solving time limit to 1,000s. The results show that data augmentation can improve the performance.

Table 17: The effects of data augmentation on solving performance.

|  | Obj ↑ | $\text{gap}_{abs}$ ↓ |
|---|---|---|
| w/o data augmentation | 97393.65 | 222.94 |
| Ours | **97487.18** | **129.41** |

## H.3 COMPARISON WITH MORE BASELINES

**Comparison with Different Fixing Strategies**    We provide more results when comparing different fixing strategies on more benchmarks. We set the search time in each iteration to be consistent across these methods. Direct Fixing relies totally on the GNN predictor for variable fixing, and Multi-stage PS relies on the reference solution provided by the traditional solver. Different from these two baselines, our proposed method introduces the correction mechanism and combines the predicted and reference solutions, determining the most confident variables to fix. The results are presented in Table 18. The results in Table 18 show that our proposed prediction-correction method outperforms the other baselines.

Table 18: Comparison of solving performance between our approach and different fixing strategies, under a $1,000$s time limit. We report the average best objective values and absolute primal gap.

|  | CA (BKS 97616.59) | | SC (BKS 122.95) | | IP (BKS 8.90) | | WA (BKS 704.88) | |
|---|---|---|---|---|---|---|---|---|
|  | Obj ↑ | $\text{gap}_{abs}$ ↓ | Obj ↓ | $\text{gap}_{abs}$ ↓ | Obj ↓ | $\text{gap}_{abs}$ ↓ | Obj ↓ | $\text{gap}_{abs}$ ↓ |
| Gurobi | 97297.52 | 319.07 | 123.40 | 0.45 | 9.38 | 0.48 | 705.49 | 0.61 |
| PS+Gurobi | 97358.23 | 258.36 | 123.30 | 0.34 | 9.17 | 0.27 | 705.45 | 0.57 |
| Direct Fixing+Gurobi | 96939.19 | 677.4 | 123.30 | 0.35 | 9.22 | 0.32 | 705.40 | 0.52 |
| Multi-stage PS+Gurobi | 97016.47 | 600.12 | 123.20 | 0.25 | 9.18 | 0.28 | 705.33 | 0.45 |
| Ours+Gurobi | **97487.18** | **129.41** | **123.05** | **0.10** | **8.90** | **0.00** | **704.88** | **0.00** |

**Comparison with a Warm-starting Gurobi**    We provide more results when comparing the warm-starting methods on more benchmarks. The results are presented in Table 19, in which we set the solving time limit as 1,000s.

## H.4 EVALUATION ON MORE BENCHMARKS

We evaluate our method in more benchmarks, including the small-scale CA dataset, the large-scale and real-world dataset SCUC, and real-world general integer programming problems APS. We report the experiment results in Table 20. The experiment results demonstrate the strong performance of our proposed method across various benchmarks. Notice that our method still outperforms the baselines in the general integer programming problems (APS) in Table 20.

Table 19: Comparison of solving performance between our approach and the warm-starting methods, under a $1,000$s time limit. We report the average best objective values and absolute primal gap.

| | CA (BKS 97616.59) | | SC (BKS 122.95) | | IP (BKS 8.90) | | WA (BKS 704.88) | |
|---|---|---|---|---|---|---|---|---|
| | Obj ↑ | gap$_{abs}$ ↓ | Obj ↓ | gap$_{abs}$ ↓ | Obj ↓ | gap$_{abs}$ ↓ | Obj ↓ | gap$_{abs}$ ↓ |
| Gurobi | 97297.52 | 319.07 | 123.40 | 0.45 | 9.38 | 0.48 | 705.49 | 0.61 |
| PS+Gurobi | 97358.23 | 258.36 | 123.30 | 0.35 | 9.17 | 0.27 | 705.45 | 0.57 |
| WS-PS+Gurobi | 97016.34 | 600.25 | 123.30 | 0.35 | 9.20 | 0.30 | 705.45 | 0.57 |
| WS-Ours+Gurobi | 97359.19 | 257.40 | 123.20 | 0.25 | 9.13 | 0.23 | 705.40 | 0.52 |
| Ours+Gurobi | **97487.18** | **129.41** | **123.05** | **0.10** | **8.90** | **0.00** | **704.88** | **0.00** |

Table 20: Comparison of solving performance on more benchmarks, under a $1,000$s time limit. We report the average best objective values and absolute primal gap.

| | SCUC (BKS 1254399.66) | | Smaller-Size CA | | APS (BKS 558917.52) | |
|---|---|---|---|---|---|---|
| | Obj ↓ | gap$_{abs}$ ↓ | Time ↓ | gap$_{abs}$ ↓ | Obj ↓ | gap$_{abs}$ ↓ |
| Gurobi | 1269353.86 | 14954.20 | 105.61 | 0.00 | 666583.20 | 107665.68 |
| ND+Gurobi | 1266355.77 | 11956.11 | 102.34 | 0.00 | 646908.07 | 87990.55 |
| PS+Gurobi | 1265332.21 | 10932.55 | 104.68 | 0.00 | 635087.43 | 76169.91 |
| ConPS+Gurobi | 1264173.98 | 9774.32 | 98.63 | 0.00 | 626713.56 | 67796.04 |
| Ours+Gurobi | **1261684.89** | **7285.23** | **94.04** | **0.00** | **603443.23** | **44525.71** |

## H.5 GENERALIZATION

We evaluate the generalization performance of our method. Specifically, we generate larger instances of the CA and SC problems (please refer to Appendix F.2 for more details). We utilize the model trained on the instances described in Section 5.1 and evaluate the models on these larger instances. The experiment results in Table 21 demonstrate the strong generalization ability of Apollo-MILP, as it outperforms other baselines on these larger instances.

Table 21: We evaluate the generalization performance on 100 larger instances. The ML approaches are implemented using Gurobi, with a time limit set to 1,000s. '↑' indicates that higher is better, and '↓' indicates that lower is better. We mark the **best values** in bold.

| | CA (BKS 115746.88) | | SC (BKS 101.45) | |
|---|---|---|---|---|
| | Obj ↑ | gap$_{abs}$ ↓ | Obj ↓ | gap$_{abs}$ ↓ |
| Gurobi | 114960.25 | 786.63 | 102.29 | 0.84 |
| ND+Gurobi | 115035.44 | 711.44 | 102.51 | 1.06 |
| PS+Gurobi | 115228.20 | 518.68 | 102.27 | 0.82 |
| ConPS+Gurobi | 115343.23 | 403.65 | 102.18 | 0.73 |
| Ours+Gurobi | **115413.86** | **333.02** | **102.16** | **0.71** |

# I  REPRODUCTION OF THE BASELINES

Since ND and ConPS are not open-source, we must reproduce the results from the original papers to validate our models. In this section, we reproduce the experiments conducted in the original studies (Nair et al., 2020) and (Huang et al., 2024) to ensure the performance of our reproduced models.

Following the validation approach and settings outlined in Nair et al. (2020) and Han et al. (2023), we conduct experiments on the Neural Network Verification (NNV) dataset. We implemented the Neural Diving method within SCIP and compared its performance against the default SCIP. As noted by Han et al. (2023), tuning the presolve option in SCIP can lead to false assertions of feasibility in the NNV dataset; therefore, we disabled this option for both SCIP and our reproduced ND+SCIP. The results are presented in Figure 4, where ND significantly outperforms SCIP, confirming the performance of our reproduced model.

To reproduce ConPS, we utilized the IP dataset built on SCIP and replicated the experiments described in the original paper (Huang et al., 2024). The results are summarized in Figure 5, where the performance of ConPS aligns with the results in the original study (Huang et al., 2024).

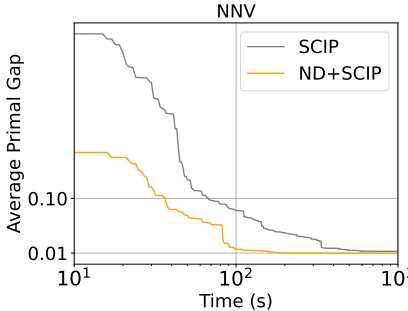

Figure 4: The reproduced results of SCIP and ND+SCIP on the NNV dataset.

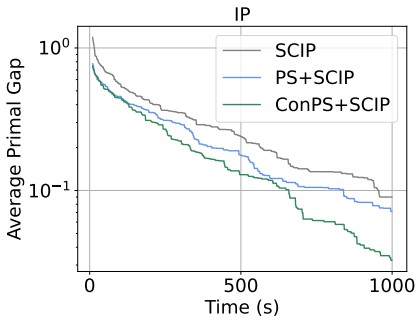

Figure 5: The reproduced results of SCIP and ConPS+SCIP on the IP dataset.

