# OpenReview forum: "Apollo-MILP: An Alternating Prediction-Correction Neural Solving Framework for Mixed-Integer Linear Programming"
_ICLR.cc/2025/Conference — ICLR 2025 Poster_

### Official Review · Reviewer_CDjs · 2024-10-27

**Soundness:** 2
**Presentation:** 2
**Contribution:** 2
**Rating:** 5
**Confidence:** 5

**Summary:**

The authors propose a Prediction-Correction Mixed-Integer Linear Programming (MILP) solving framework, referred to as Apollo-MILP, aimed at addressing MILP problems. In each iteration, Apollo-MILP first predicts a partial solution that will be fixed and subsequently improves this solution using a trust region method. Under a set of strong assumptions, theoretical results indicate that Apollo-MILP ensures improved solution quality and feasibility. Furthermore, experimental evaluations **seem to** demonstrate that the proposed framework outperforms the state-of-the-art methods across several datasets. **Generally speaking, I think this paper is a trivial extension of the work [3], and I personally do not see significant contributions**.

**Strengths:**

I think the proposed Apollo-MILP framework is new, and the iterative predict-based fixing strategy is well designed for MILPs. Moreover, the perspective of de-coulping  the fixing strategy into two steps--prediction and correction--is interesting.

**Weaknesses:**

I have a few concerns below.

- Generally, this paper presents some theorems under several very strong assumptions. I do not think this paper has theoretical contribution, it is more about presenting an empirical algorithm. For example, in Theorem 4, "If the trust-region searching problem 2 is feasible....", the theorem is clearly valid if this assumption is true. I do not think this is worth a theorem.

- Assumption 1 is kind of too strong and counter-intuitive, can you provide some experiments for validation?

- To compare the proposed method against ND [1] and ConsPS [2], I believe it is essential to reproduce some of the experiments presented in their original papers to ensure that ND and ConsPS  are indeed comparable.

- In Table 6, we notice that the parameters $k_0$, $k_1$, and $\Delta $ differ significantly from those used in PS's paper [3]. Why did you choose not to use the original parameters? Parameters can directly influence the performance of both the PS and the fixing strategy. For a fair comparison, we recommend the authors employ the optimal parameters for your baseline methods.

- To clearly demonstrate the performance of Apollo-MILP, I suggest including at least two experiments in your ablation study.
  * Case i: Apollo-MILP v.s. 4 rounds of Direct Fixing;
  * Case ii: Apollo-MILP v.s. 4 rounds of PS;

   All these methods utilize the same parameters $k_0^{(i)}, k_1^{(i)}$ and $\Delta^{(i)}$, except in the case of Direct Fixing, where $\Delta$ is not present. In each iteration, the solving times are consistent across these methods. By conducting these two experiments, you can effectively demonstrate that the performance improvements result from Apollo-MILP's multi-stage correction rather than from direct multi-stage solving (e.g., 4 rounds of PS and 4 rounds of Direct Fixing). Additionally, this approach will highlight the effectiveness of combining prediction and correction within your framework.

Minor Issues:

- typo: page 6, paragraph 2. “Instead, UEBO offers a practical estimation by utilizing the available distributions $p_θ(x_i | I)$, $q(x | \hat x_i, I)$”,  $q(x | \hat x_i, I)$ is missing an $i$.

- The WA's figure is missing in Fig.3

[1] Vinod Nair, Sergey Bartunov, Felix Gimeno, Ingrid Von Glehn, Pawel Lichocki, Ivan Lobov, Bren- dan O’Donoghue, Nicolas Sonnerat, Christian Tjandraatmadja, Pengming Wang, et al. Solving mixed integer programs using neural networks. *arXiv preprint arXiv:2012.13349*, 2020.

[2] Taoan Huang, Aaron M Ferber, Arman Zharmagambetov, Yuandong Tian, and Bistra Dilkina. Con- trastive predict-and-search for mixed integer linear programs. In Ruslan Salakhutdinov, Zico Kolter, Katherine Heller, Adrian Weller, Nuria Oliver, Jonathan Scarlett, and Felix Berkenkamp (eds.), *Proceedings of the 41st International Conference on Machine Learning*, volume 235 of *Proceedings of Machine Learning Research*, pp. 19757–19771. PMLR, 21–27 Jul 2024. URL https://proceedings.mlr.press/v235/huang24f.html.

[3] Qingyu Han, Linxin Yang, Qian Chen, Xiang Zhou, Dong Zhang, Akang Wang, Ruoyu Sun, and Xi- aodong Luo. A gnn-guided predict-and-search framework for mixed-integer linear programming. In *The Eleventh International Conference on Learning Representations*, 2023

**Questions:**

See the weakness part.

---

### Official Review · Reviewer_PGi6 · 2024-10-30

**Soundness:** 4
**Presentation:** 3
**Contribution:** 3
**Rating:** 8
**Confidence:** 3

**Summary:**

This paper introduces Apollo-MILP, a novel iterative framework for MILP that combines prediction and correction. The core idea involves alternating between prediction and correction steps to reduce search space and improve the solution: A bipartite GNN predicts a solution, and a better solution is further searched in the trust region. Another contribution is UEBO, a theoretical sound upper bound, which evaluates the reliability of the current solution and guides variable fixing to reduce problem size after correction. The framework demonstrates superior performance over traditional solvers (Gurobi & SCIP) and other learning methods on various benchmark datasets, achieving comparable or better solutions within a shorter time frame.

**Strengths:**

1. **Novelty:** Apollo-MILP introduces a novel approach to MILP by combining gin-based prediction with an iterative correction process. This reduces problem dimensionality and maintains solution quality through each iteration. The Uncertainty-based Error Upper Bound (UEBO) is also a good contribution, measuring prediction reliability to guide variable fixing.
2. **Effectiveness:** The experimental setup and result quality are good, with well-chosen benchmarks and comparisons to MILP solvers and other SOTA baselines. The experiments clearly demonstrate its advantages in speed and accuracy.
3. **Clarity:** The paper is well-written and well-structured, and its explanations are generally clear, with enough detail provided for the algorithm, theoretical foundations, and experimental setup.

**Weaknesses:**

1. **Time-Consuming Data Collection:** For my understanding, a key limitation is the time-consuming nature of data collection. Since the model requires optimal or near-optimal solutions as labels during training, it depends on solving many problem instances, which can be computationally expensive for large problems.
2. **Limited Benchmark Problems:** The number of benchmarks is 4, which, while indicative, may limit the generalizability of the results. Although it is understandable that solving large-scale MILPs can be time-consuming, expanding the experimental scope to include more problems, even some smaller-scale instances, would provide a more comprehensive view.

**Questions:**

1. Could the authors provide further details on the characteristics of the problem instances used in the experiments? Specifically, information on decision variables, such as the number of continuous variables, integer variables, and binary variables, would be valuable.
2. The authors claim that Apollo-MILP suits general integer programming (IP) problems. I wonder now if there are any experimental results on general IP problems, not just 0-1 variables.
3. The current approach appears to fix one variable with the lowest UEBO in each iteration. Has the team considered fixing multiple variables with low UEBO values simultaneously? Testing this multi-variable fixing strategy could potentially enhance the convergence rate by accelerating problem reduction.

---

### Official Review · Reviewer_NfYM · 2024-11-03

**Soundness:** 3
**Presentation:** 2
**Contribution:** 3
**Rating:** 6
**Confidence:** 3

**Summary:**

This paper proposes a framework, Apollo-MILP, to obtain solutions for MILPs. The framework combines a prediction phase with a correction phase in an iterative manner. In each iteration, the predictor estimates values for unfixed variables, followed by a correction step that refines the partial solution. The correction step is analysed with a newly introduced uncertainty-based error upper bound to evaluate the reliability of predictors, which helps determine which variables to confidently fix. This alternating framework allows the system to reduce problem size iteratively while preserving solution quality and feasibility.Benchmarking results show that Apollo-MILP outperforms both ML-based and traditional solvers, achieving better objective values and closer gaps to the best-known solutions.

**Strengths:**

1. The paper introduces a new Uncertainty-based Error upper Bound (UEBO) metric and provides a detailed theoretical analysis of the proposed framework.

2. The paper demonstrates Apollo-MILP's effectiveness across various benchmark datasets, showing improvements in solution quality and convergence speed compared to both traditional and ML-based solvers.

**Weaknesses:**

1. The presentation of the paper can be improved with better introduction of the background and basic concepts. For instance, Proposition 1 uses the concepts of entropy and KL divergence, which should be defined in the context of solution prediction for MILPs.

2. The general framework would be clearer if Algorithm 1 from Appendix B were incorporated into the main text alongside the description in Section 4. Moving the algorithm into the main content could improve readability and help readers follow the approach more easily.

**Questions:**

1. The proposed alternating problem-solving and variable-fixing strategy resembles large neighborhood search (LNS). In each iteration, the ML model predicts a "confidence" level for fixing a variable to either 0 or 1, and the top k0+k1 variables are fixed to construct the reduced problem in Equation (2). A MILP solver then solves this reduced problem to obtain a “reference solution.” Before the next iteration, variables with shared values between the “predicted solution” and “reference solution” are fixed. The ML model effectively selects the neighborhood for LNS. Could you please clarify the distinctions between the proposed framework and ML-based LNS methods in the literature?

2. Could you please comment on the rationale for some design choices of the implementation? For instance,
why is the algorithm divided into four iterations with 100, 100, 200, and 600 seconds respectively? What are the other alternatives that have been experimented?

3. In Appendix F, hyperparameters are given for the framework. What is the search space for hyperparameters?

---

### Official Review · Reviewer_Gah2 · 2024-11-11

**Soundness:** 3
**Presentation:** 3
**Contribution:** 3
**Rating:** 6
**Confidence:** 5

**Summary:**

In the context of deep learning for solving mixed-integer linear programs (MILPs), this paper proposes enhancements to an existing predict-and-search framework.

Training differs from existing work in that the supervised learning dataset includes not only near-optimal solutions to training MILP instances, but also to sub-instances (restrictions) in which some variables are fixed and only other variables are optimized. A GNN is trained to maximize an energy potential function that assigns higher probabilities to variable value-assignments that have a better objective function value.

Most proposed changes are at inference time. First, the approach is iterative. In every iteration, a subset of variables have been fixed based on preceding iterations. Then, the GNN makes a prediction on the remaining variables. A trust-region search is performed around said solution using a MILP solver for a limited amount of time. The GNN prediction and the trust-region solution are compared using an upper bound on GNN prediction uncertainty. Based on this bound, some of the GNN’s variable assignments are ignored if too uncertain. The newly fixed variables are appended to those from the previous iteration and the next iteration of this prediction-correction loop is performed until a set number of iterations is met.

Experiments on four MILP benchmarks from the literature show that the proposed framework produces better solutions on average than Gurobi and three comparable ML methods from the literature. Ablations and hyperparameter searches are performed to justify certain elements of the proposed method and tune its performance to a validation set.

**Strengths:**

S1. The incorporation of uncertainty quantification is new and interesting in this setting. Although I have clarification questions on this (see Weaknesses), the overall approach seems sensible and effective in practice. Some theoretical results justifying the proposed uncertainty bound, UEBO, are also presented.

S2. The experiments span multiple competing methods, four very reasonable MILP benchmarks, a subset of the MIPLIB2017 collection, ablations, and hyperparameter analysis. Although I have questions on other aspects of the results in the paper, I appreciate the care taken in designing the experiments.

**Weaknesses:**

W1. Table 8 says that the CA instances have 1500 variables and ~2500 constraints. The “Hard” instances of CA from Gasse et al. Have 1500 bids, which I believe corresponds to 1500 decision variables. However, their Table 2 “Hard” column for CA shows that SCIP with RPB branching (the default in SCIP) has an average running time of 136.17 seconds, meaning all or most instances are solved to global optimality in 2.5 minutes on average. In contrast, your Table 1 for CA shows that Gurobi typically terminates with a suboptimal solution in 1000 seconds. This is extremely puzzling to me as Gurobi is faster than SCIP and you are likely using better CPUs than used in Gasse et al. Could you clarify whether your CA instances in Table 1 correspond indeed to the “Hard CA” instances of Gasse et al.? Could you explain the performance discrepancy? These instances seem to be super easy to solve to optimality; why does this not apply in your experiments?

W2. Please clarify how you calculate the UEBO, particularly the second term. I have struggled to interpret the conditional version of the q(.) distribution that seems so crucial to this bound. Perhaps a small numerical example is in order for this rebuttal and updated versions of this paper.

W3. Solvers such as Gurobi can be used to emphasize finding good solutions quickly. In particular, MIPFocus=1 (https://www.gurobi.com/documentation/current/refman/mipfocus.html). Do you use this for the “Gurobi” row of your results Table 1? If not, then you should as it is unfair to compare to default Gurobi on primal gap when it is running to balance between proving optimality and finding solutions. It should focus on the latter.

W4. Warm-starting Gurobi as baseline: What if you passed the initial GNN prediction (fixing the k0 and k1 variables to 0 and 1 and letter other variables free) to Gurobi as a MIPStart (https://support.gurobi.com/hc/en-us/articles/360043834831-How-do-I-use-MIP-starts)? Gurobi can search around this warm-start and various ways and it has parameters that let you control this process. I believe this baseline is crucial to understanding the benefits of your and other similar methods. The methods you compare to may have missed this, but that does not justify you not performing this experiment.

**Questions:**

Kindly refer to the weaknesses. I can be convinced to increase the rating if convincing answers are provided.

Minor comments:
- line 256: the notation for equality of two variable assignments along index set P can be confusing. It is better to say x_P[P] = x[P]. Set membership with $\in$ has a very specific meaning that is not met by your definition.

---

### Meta-Review · Area_Chair_Fn64 · 2024-12-22

**Metareview:**

This paper proposed an alternating prediction and correction method to solve mixed-integer linear programs (MILP). In each iteration, it predicts some variables for reoptimization, followed by a correction model to refine the prediction. It also has an uncertainty-aware design to fix variables of high confidence. Reviewers agree that the prediction-correction design and the incorporation of uncertainty quantification are novel ideas, and the empirical performance is strong. They also pointed out several limitations, including 1) presentation could be improved; 2) computationally demanding in collecting labels since MILP is NP-hard; and 3) assumptions are too strong, making the theoretical analysis less practical. After rebuttal, most of the concerns are addressed, and three out of four reviewers are positive. I agree with the reviewers that the contribution of this paper is interesting, and makes non-trivial progress in neural MILP solving. I recommend acceptance, and urge the authors to incorporate all comments in the final paper, especially the discussion on the assumptions. Also, please release your code for better reproduction.

**Additional Comments On Reviewer Discussion:**

Authors' responses addressed most of the concerns, and two reviewers increased their score. After reading all replies, I think authors did a good job in responding to the reviewers' comments.

---

### Decision · Program_Chairs · 2025-01-22

Accept (Poster)